

# 1 Imaging of the active root current pathway under partial root-zone drying

# 2 stress: A laboratory study for *Vitis vinifera*.

Benjamin Mary[1,2], Veronika Iván[1], Franco Meggio[3,4], Luca Peruzzo[1,2], Guillaume Blanchy[5], Chunwei
Chou[2], Benedetto Ruperti[3,4], Yuxin Wu[2], Giorgio Cassiani[1]
[1]Dipartimento di Geoscienze, Università degli Studi di Padova, Padova, Italy
[2]Earth and Environmental Sciences Area, Lawrence Berkeley National Laboratory, California, USA
[3]Department of Agronomy, Food, Natural resources, Animals and Environment – DAFNAE, University of Padova, Agripolis,
Viale dell'Università 16 – Legnaro (Padova), Italy;
[4]Interdepartmental Research Centre for Viticulture and Enology - CIRVE, University of Padova, Via XXVIII Aprile 14,
Conegliano (Treviso), Italy;
[5]Urban and Environmental Engineering, University of Liège (ULiege), Liège, Belgium
*Correspondence to*: B. Mary (benjamin.mary@unipd.it)
**Abstract**
Understanding root signals and their consequences on the whole plant physiology is one of the keys to tackling the water-
saving challenge in agriculture. The partial root-zone drying (PRD) method is part of an ensemble of irrigation strategies that
aim at improving water use efficiency. To reach this goal tools are needed for the evaluation of the root's and soil water
dynamics in time and space. In controlled laboratory conditions, using a rhizotron built for geoelectrical tomography imaging,
we monitored the spatio-temporal changes in soil electrical resistivity for more than a month corresponding to six Partial
Rootzone Drying (PRD) cycles. Electrical Resistivity Tomography (ERT) was complemented with Electrical Current Imaging
(ECI) using plant stem-induced electrical stimulation. We demonstrated that under mild water stress conditions, it is practically
impossible to spatially distinguish the PRD effects using ECI. We evidenced that the Current Source leakage depth varied
during the course of the experiment but without any significant relationship to the soil water content changes or transpiration





demand. On the other hand, ERT showed spatial patterns associated with irrigation and, to a lesser degree, to RWU. The
interpretation of the geoelectrical imaging with respect to root activity was strengthened and correlated with indirect
observations of the plant transpiration using a weight monitoring lysimeter and direct observation of the plant leaf gas
exchanges.

## 1. Introduction

In the context of water scarcity, agriculture needs to improve irrigation practices by reducing water inputs and selecting
adequate species and, in the case of woody crops, most efficient scion-rootstock combinations. In order to evaluate the efficacy
of irrigation, it is necessary to develop tools capable of evaluating root functioning and quantifying root water uptake. The
partial root zone drying (PRD) method is part of an ensemble of irrigation strategies that aim at improving water use efficiency.
It consists of irrigating only one part of the root system of the same plant using a certain percentage of the potential
evapotranspiration (ETp), usually inferior to the total water needed. Application of PRD triggers a physiological response in
the plant via a hormone called Abscisic acid (ABA), which is produced in the roots and transmitted to the leaves to regulate
the stomata closure and thus reducing water transpiration while keeping photosynthesis active and finally leading to increased
water use efficiency. A number of publications investigated the origins of the mechanism controlling transpiration during PRD
(Stoll et al., 2000), while others focused on the consequences in terms of  Root Water Uptake (RWU) and production crop
yield (Collins et al., 2009).
The plant's natural bioelectrical activity is necessary for its physiological processes. Plant scientists represent it by a water
column where the ions move from bottom to top and vice versa due to gradients of water potentials. In their studies, Voytek
et al. (2019) and Gibert et al. (2006) successfully linked the measurements of electrical potential in the ground and in the tree
stem to the RWU and sap flow respectively.  The use of active methods such as electrical resistivity tomography (ERT) allows
for spatial and temporal analysis of the subsoil. Recent advances in electrical tomography imaging, in particular reduced at the
plant scale, show their effectiveness to measure changes in soil water content associated with the RWU (e.g. Cassiani et al.,
2015, 2016; Mary et al., 2018). Applications of geoelectrical methods to evaluate water use efficiency are increasing. Recently



in an experimental Citrus orchard, Consoli et al., (2017), Vanella et al., 2018 and Mary et al., (2019a) showed that soil moisture
patterns determined by PRD are visible from the ERT perspective and can be attributed to the root system distribution.
However, processes occurring in the rhizosphere can affect the soil electrical resistivity (ER) in various ways. Roots induce
changes in the soil structure in terms of porosity and hydraulic conductivity which ultimately modify the water pathways and
fluxes and thus the ER itself. Stemflow channelling by roots is an example of how water from rain or irrigation can be driven
to soil recharge by the root structure. Conversely, root uplift in agroforestry shows how water can move from the deeper layers
to the top via the roots.
Roots also affect the soil ER through the geochemical changes associated with root exudates and root symbiosis. At the
interface between soil and roots, the chemical gradients and concentrations can drastically differ from those observed in the
soil regions not affected by the roots. Although this can have a significant impact and be a valuable source of information,
only a few studies have extended the ERT and the induced polarisation (IP) to observe these changes (Weigand, 2017; Weigand
and Kemna, 2019; Tsukanov and Schwartz, 2020, 2021). As of today, the electrical behaviour of individual roots remains
poorly understood, particularly with regard to their changes in type (from hair roots to fully lignified roots), space, time, and
whether the root is active or not (Ehosioke et al., 2020).
The geophysical approach extends the scope of traditional methods to evaluate soil water content (SWC) using time-domain
reflectometry (TDR) sensors and the calculation of RWU (Jackisch et al., 2020). In the field, the spatial resolution is controlled
(in ERT or IP) by the arrangement of the electrodes and acquisition parameters (Uhlemann et al., 2018), while the temporal
resolution is controlled by the time it takes to complete a full sequence measurement.
Rhizotrons are one of the earliest and most effective tools for studying root growth and functioning, both in the field and in
the laboratory (Taylor et al., 1990). They are transparent boxes that allow the direct observation of the roots during plant
growth and changes in soil conditions. Rhizotrons also provide valuable support in multidisciplinary studies, allowing other
methods to be more easily and precisely deployed, so that their results more reliably interpreted. For example, a load scale is
often mounted in combination with the rhizotron in order to weigh the system, which allows inferring the quantity of water
lost by the plant over time. This set-up is inspired by the lysimeter and is widely adopted to measure the water balance of the





soil-plant interactions. For example, in a rhizotron, Doussan and Garrigues (2019) use the light transmission 2D technique to
infer root water uptake with respect to their genotypes.
The very few studies conducting geophysical tomography imaging in the laboratory using a rhizotron proved a certain
efficiency in studying the interaction between soil physics and plant physiology for predicting plant response to environmental
stresses (Weigand, 2017, 2019; Peruzzo et al., 2020). It allows for high-resolution tomography by reducing the size, diameter,
and spacing of the electrodes. The entire soil profile is easily accessible by placing electrodes on the side of the rhizotron,
easing the depth resolution limitation inherent to surface-based geophysical methods usually used for field acquisition.
Although there is a good momentum for the use of geophysical methods applied to agronomy (Garré et al., 2021), a number
of gaps still need to be addressed. All the indirect root effects on the soil ER affect the evaluation of the soil water content,
making the interpretation of ERT to quantify RWU sometimes difficult (Ehosioke et al., 2020).

### 1.1.   Current pathways in roots under PRD constraints

Current pathways in roots remain certainly the main unknown since there is a gap in techniques to measure

it non-destructively (Ehosioke et al., 2020; Liu et al., 2021). The current pathways in roots are possibly

linked to RWU. Lovisolo et al. (2016) describe in detail the flow of water from root water uptake and the

processes occurring at the cell scale. In any case, root water uptake is not distributed equally over the whole

root system, due to in part of heterogeneous soil conditions. For the same reason as soil saturation can change

over time, RWU is also varying in the time. For active roots, root water uptake consists in a moving water

from the root tip (which is usually much more electrically conductive due to high water conductivity at its

proximity) in the radial direction via cellular (symplastic way) and between cells (appoplastic way) until it

reaches the xylem which transport it in the axial direction towards the upper part. Water flow can encounter

resistances due to suberization (conversion of the cell walls into cork tissue by development of suberin),

which is naturally driven as a consequence of root growth (secondary roots are more suberised than primary

roots) but it can also be the consequence of plant stress (Malavasi et al., 2016; Song et al., 2019). The process

can cause reductions in water conductivity through the root system by limiting the permeability of the root

tissue, thus leading to changes in the plant's ability to take up water. For the specific PRD case, there is a





complex balance between reducing radial flow (as a consequence of ABA signalling sent by the roots) to
conserve water in the soil but keeping the axial flow active. This can be done for instance by adjusting the
xylem vessels size and quantities. Although suberisation is usually a long-term process, studies show that
PRD can promote and accelerate the process of suberization in response to water limitation. Finally during
PRD conditions we can also observe transfer of water from the wet to the dry side through the roots
(overnight) in a process called redistribution (Yan et al., 2020), which induces spatio-temporal variations in
RWU that ultimately influences also electrical current pathways in roots.
A direct approach to analysing the active part of the root system consists of an injection of current stimuli
into the plant stem. The so-called "capacitance approach" has been developed for years by plant
physiologists, starting from the theory developed by Dalton (1995) who conceptualized the current pathways
through the root xylem by an equivalent parallel resistance-capacitance circuit. The theory holds under the
assumption that the current flows throughout the most conductive path and is held (thus inducing
polarization) by the root cell membranes before being released into the soil. Since then, contrasted
experimental results opposed on the relationship between root capacitance ("ECroot") and root traits in
various crops, particularly because of studies supporting the major contribution of the stem compared to the
roots on the total ECroot measured and the possible current leakage at the proximal part (Urban et al., 2011;
Dietrich et al., 2018; Peruzzo et al., 2020).
Without being able yet to give hints about the electrical current pathway, recent advancements in the
development of explicit RWU models, based on plant hydraulics, provide insights into how robust
capacitance models hold and under which conditions. We learnt, for instance, that at the root level, RWU
models account for the anisotropy by separating the root hydraulic conductance into two terms (longitudinal
and radial). The same applies to the stem-based methods as root hydraulic conductance and electrical
conductivity are likely to vary conjointly. Up to now the relationship between root water content and root
hydraulic conductivity with electrical resistivity has not been firmly established. Many other parameters can
affect the water flow as well as the current pathway of stem-based methods.



Peruzzo et al. (2020) hypothesize that drought stress can also reduce electrical current leakage, particularly
for woody species. Furthermore, as expected, the frequency of the injected current plays an important role
in the capacitance measured. At high frequencies, both the longitudinal conductivity and radial conductivity
increase (Mancuso 2012; Ehosioke et  al. 2020), which can also cause current leakage problems (Gu et al.,
2021). The measure of plant responses over multiple frequencies, a method called Electrical Impedance
Spectroscopy (EIS) is more time-consuming but more informative since different polarisation processes can
manifest themselves in the signal (Ehosioke et al., 2020). The contrast of electrical resistivities between soil
and roots plays a fundamental role as reported e.g. by Cseresnyés et al. (2020). Gu et al. (2021) stated that
the potential to directly quantify root traits under dry conditions is higher than under wet conditions and
interpreted this as a result of the fact that the root electrical longitudinal conductivity is higher than that of
the soil under dry conditions. The instrumentation and acquisition schemes used for EC are also questionable
and the optimal experimental setup of measurement remains to be determined (Postic and Doussan, 2016).
The number and the position of the stem and the return electrodes are a cause of uncertainties (electrode
contact resistance, etc.). Peruzzo et al. (2021), in a three channels experiment, were able to provide direct
access to the response of stem and soil, which ultimately allowed the decoupling of the root response.
Evidence showed the presence of current leakage in herbaceous root systems, a significant contribution from
plant stem, and a minor impact from the soil.
Gu et al. (2021) stated that in addition to the traditional regression model used for predicting root traits using
the EC method, a forward model would help to illustrate the importance of these different factors. In order
to cope with the main drawbacks of the EC methods, we propose the so-called Electrical Current Imaging
(ECI) method, a physically based approach based on recovering the current density distribution instead of
simply calculating the total resistance/capacitance. This method is also referred to as mise-à-la-masse
(MALM) in the applied geophysics literature. The current imaging methods hold some promise to offer a
first set of evidence about the current pathways: This is a popular technique adopted e.g. by the neurosciences
community, where the current density in the human brain correlates with diverse patterns of neural activity





(Kamarajan et al., 2015). Peruzzo et al. (2020) applied it for plant roots imaging with relative success, as the

authors stated that all the current leaks at the plant's proximal part i.e. at the shallowest contact of the plant

stem with the soil. For the ECI approach, the Poisson's equation serves as a physical model for the electrical

current flow. As current flow is modulated by the conductivity of the soil, the ECI approach is always

combined with ERT  in order to recover of the soil resistivity distribution.

**1.2.    Study aims and assumptions**

The aim of this study is twofold:

(i) we aim at showing that the current path through the root system is linked to the active root zones.

(ii) we want to investigate how the soil water content affects the current path.

For this, we rely on the following assumptions:

- changes in soil water content measured by ERT are a relevant spatial

proxy of root activity and can be used as an indicator of the actual plant

transpiration by correlating them with variations of the total rhizotron

measured weight.

- during the application of PRD, only one part of the root system would

be active and the current injected in the stem would preferably spread

to the side where the root system is irrigated.




**2.    Material and methods**



## 2.1. Experimental setup

### 2.1.1. Rhizotron

The experiment was conducted using a rhizotron 50 cm wide, 50 cm high, and 3 cm thick, with a transparent screening face. The front of the rhizotron was equipped with 64 stainless steel electrodes with 4 mm diameter which did not extend into the rhizotron's inner volume (Fig. 1). An additional line on the top surface of the rhizotron was composed of 8 electrodes inserted to 1 cm depth. A growth lamp was installed above the rhizotron and turned on during daylight hours (from 7 am to 7 pm). The rhizotron was closed on all sides and watertight, with only 8 small holes used for the irrigation at the surface and the central hole where the plant is placed. We considered the surface of these holes to be sufficiently small to neglect the possible effect of evaporation through them. An outlet point was placed on the bottom right side (z=5cm) and the rhizotron was always saturated below this point. In the course of the experiment (after the growing period) no water discharge was observed through the outlet point.



**Figure 1: Conceptual figure showing the position of the plant in the rhizotron. The water input was done alternatively from left (a) to right (b) via small holes on the top of the rhizotron (H1 to H8). The roots are free to grow on both sides of the rhizotron. The circles on the screening face show the locations of the electrodes. Two additional electrodes (needles) are used for the ECI, one for the stem injection and the other for the control soil injection next to the stem. The rhizotron is weighted by a central point load scale (PC60-30KG-C3, Flintec) mounted between two support plates in plexiglass.**

### 2.1.2. Plant treatment

At the initial stage of the experiment, we used a *Vitis Vinifera* cutting with a pre-developed root system (rooted cutting var. Merlot) was used. The cutting was grown in hydroponic solution (modified Hoagland medium) for 4 months before being transferred into the rhizotron. This was followed by a growing period of 5 weeks with irrigation applied over the whole width of the rhizotron every 3 days. The vine was then irrigated with a nutrient solution (see Table 1) following a PRD protocol.



### 2.1.3. Soil type

The experiment was conducted in a sand-peat mixture (50-50 m/m%). The applied sand was high-purity quartz sand ($SiO_2$ = 99%) of grain size comprised between 0.1-0.6 mm and the peat was a normal commercial acidic sphagnum peat. During the course of the experiment, the soil was stable through time with very low compaction (1 cm) observed at the end of the experiment (already observed by Doussan & Garrigues, (2019) for soil with a lower density than 1.5-1.6 g/cm$^3$). The sand-peat mixture was chosen as a compromise between water retention and drainage. We estimated the porosity at the beginning of the experiment as equal to 55% using the ratio of water weight after saturation to the total volume of the rhizotron.

### 2.1.4. Irrigation schedule

For each irrigation we regulated the amount of water supplied based on the information obtained from the scale data, the plant received 75% of the measured transpiration. For each cycle, the wetting side changed (from left to right). Note that in this experiment, we did not consider a physical barrier to separate the two sides of the rhizotrons to a split-roots configuration as is the case for other PRD experiments conducted in the laboratory (Martin-Vertedor and Dodd, 2011; Sartoni et al., 2015). In general, the use of physical barriers in Partial Root Zone Drying (PRD) experiments is not always a standard aspect of the setup.

Table 1 describes all cycles conducted from April 13to July 07:

- The goal of Cycle number -1 was to ensure plant adaptation and growth after transplantation.
- Cycle numbers 0 to 2 aimed at starting the PRD irrigation with half of the rhizotron volume irrigated; i.e. we irrigated the side through a total of four holes out of eight (see Fig. 1).
- From cycle number 3 to 9, we restricted the water input only to the two lateral holes.
- Between cycles 3 and 4, we added intermediate irrigation on the full length of the rhizotron.





For the irrigation, we used a nutrient solution (modified Hoagland) (Hoagland and Arnon, 1950)
having an electrical conductivity equal to 2470±5 $\mu$S/cm (at ~25°C), except for cycle 3 where tap
water was used (560 $\mu$S/cm).


| Date (YYYY-mm-dd) | Hole (H) location (c.f. Fig. 1) | Quantity (mL)* | Cycle nb |
|---|---|---|---|
| 2022-05-13 | All | | -1 |
| 2022-05-19 | H1;H2;H3;H4 | 200 | 0 |
| 2022-05-25 | H5;H6;H7;H8 | 260 | 1 |
| 2022-05-01 | H1;H2;H3;H4 | 290 | 2 |
| 2022-06-08 | H7;H8 | 305 | 3 |
| *2022-06-10* | All | 60 | - *(3bis)* |
| 2022-06-15 | H1;H2 | 350 | 4 |
| 2022-06-22 | H7;H8 | 375 | 5 |
| 2022-06-29 | H1;H2 | 386 | 6 |
| 2022-07-05 | H7;H8 | 431 | 7 |
| 2022-07-11 | H1;H2 | 431 | 8 |
| 2022-07-12 | H1-H8 | 200 | 9 |


**Table 1: Irrigation log, indicating the date, the location where the water was input and the**
**corresponding cycle number considered in the results. Colors correspond to the side used for the**
**irrigation, green is on the right side while orange is on the left side. * Quantity in total distributed over**
**all the holes.**
**2.2.    Electrical Resistivity Tomography**
Electrical Resistivity Tomography consists in reconstructing the subsoil electrical resistivity using an array
of electrodes (Binley and Slater, 2020). In this study, a total of 72 stainless steel electrodes were used, 64





electrodes formed a grid, 5 cm spaced, covering the screening face of the rhizotron, and an additional line of
8 electrodes was posed at the top surface. Electrodes are needles 4 mm in diameter and 80 mm in length, but
only their tip is in contact with the soil. ERT involves the measurement of transfer resistances following a
sequence describing a combination of varying injections (AB) and potential (MN) pairs of the electrodes.
We used a custom sequence composed of 4968 quadrupoles including the reciprocals (e.g. Parsekian et al.,
2017),  and the measurement were conducted using a Syscal Pro (Iris Instrument) resistivity meter., The
sequence was optimized over the ten physical channels of the instrument in order to reduce the acquisition
time to approximately 30 min. The data acquisition parameters were constant along the monitoring, with a
minimum required $V_P$ of 50 mV, a maximum injection voltage $V_{AB}$ of 50 V, and a number of 3-6 stacks with
the on-time fixed to 250 ms each.

### 2.3.    Electrical Current Imaging

The electrical current imaging (or Mise-à-la-masse) method was logistically similar to ERT. The sequence
nevertheless varies, as the pairs of injection electrodes were kept constant with the positive pole (+I)
electrode located on the stem, and the return (-I) electrode located in the bottom right of the rhizotron. The
potential electrodes pairs (MN) vary according to a custom sequence. For the stem current stimulation, we
inserted a small stainless steel needle (2 cm, 1 mm diameter) into the plant stem at 5 cm from the grafted
point. The needle was inserted all the way to the centre of the stem (Fig. 1). Before each measurement, we
added a few drops of water to the stem needle in order to reduce the stem contact resistance (to values
comprised between 41 and 66 kΩ). The current was guided to the root system via the stem and then released
into the soil.
As the effect of the stem contact resistance affects the measured voltage, a control soil injection was
systematically made. In that case, the current was injected into the soil close to the plant (Fig. 1). A
qualitative comparison between the control soil injection and the stem injection plant could be made to
discriminate the effect of roots. Furthermore, soil control injection served as a visual calibration for the
inversion of the current source knowing that the injection is punctual and occurs at a known position.



### 2.4. Weight monitoring for the estimation of transpiration

In order to track the weight changes due to the transpiration of the plant, the rhizotron was equipped with a single point load cell (PC60-30KG-C3, Flintec), mounted between two plates in plexiglass supporting the rhizotron (Fig. 1). The data were logged with a sampling rate of 5 min using the weight indicator DAD-141.1. The total weight of the rhizotron is about 20 kg and the expected resolution according to the sensor datasheet is 0.1 g. The variation due to temperature was monitored, on average in May at 22°C, and in July at 25°C. To avoid sharp signal perturbation, during the irrigation and the acquisition of geophysical data the logger was paused.

### 2.5. Leaf gas exchange observations

In order to monitor the physiological response of the plant during the course of the experiment, stomatal conductance to water ($g_{sw}$ [mmol $H_2O$ m$^{-2}$ s$^{-1}$]) measurements were performed on vine leaves with an open flow-through differential porometer (LI-600, Li-Cor Inc., Lincoln, Nebraska, USA). The stomatal conductance is a measure of the density, size, and degree of opening of the stomata, therefore it can be used as an indicator of plant water status (Gimenez et al., 2005). The measurements were carried out on 26 leaves in the morning hours (at 10 a.m.), once (on 8th June 2022) just before irrigation (severe water stress), and once (on June 16, 2022) one day after irrigation (mild to low water stress). For the tracking of the plant development, the length (L) and the width (W) of every leaf were measured every 2 weeks from the beginning of the growing period until the end of the experiment. From this data the total leaf area (LA) was estimated according to three models: LA1 = 0.587 (L×W) (Tsialtas et al., 2008); LA2 = -3.01 + 0.85 (L×W) (Elsner and Jubb, 1988); LA3 = -1.41 + 0.527W$^2$ + 0.254L$^2$ (Elsner and Jubb, 1988).





**2.6.     Data processing**

**2.6.1.     Analysis of ERT data**

The ERT acquisition sequence was initially tested on the rhizotron filled with water of known

conductivity and it offered good coverage on most of the rhizotron surface with a slight decrease

on the sides. The soil electrode contact resistances varied over the course of the experiment between

5 and 20 kΩ. Data were filtered on the basis of the percentage of variations between direct and

reciprocal measurements. We chose to eliminate the data with reciprocal relative errors larger than

5%, for all the time steps. The number of rejected data varies from 9% to 39 % of the total (see

Table A1) with a median of 11%. Transfer resistances were inverted using the open-source code

ResIPy (Blanchy et al., 2020) based on the Fortran R3t code (Binley, 2015). The inversion mesh is

an unstructured grid composed of tetrahedra, created using Gmsh (Geuzaine and Remacle, 2009).

Two distinct strategies can be used: (1) individual inversion which consists of building a model of

resistivity at a given time, and (2) time-lapse inversion (difference inversion) where the difference

in resistivity is inverted between a given survey and a background survey (in this case, the

background survey is the previous one). In this study, we used the second approach, which allowed

filtering of systematic noise and highlights variations (as a percentage of differences) between two

299          times.

**2.6.2.     Analysis of current density**

The mathematical formulation for the inversion of the current source density (ICSD) has been

developed in previous studies. It consists in searching for a linear combination of Ohm's law, for a

series of current punctual sources (also called virtual sources) minimizing the misfit between

simulated and observed data. The algorithm was initially tested on the rhizotron filled with water

of known electrical conductivity and a single isolated cable (see the procedure from Peruzzo et al.,

2020). It is important to note that the ICSD relies on the knowledge of the medium conductivity (as

in the Poisson's equation, the current is modulated by the electrical conductivity). Thus, we used





the inverted ER values as the resistivity distribution for the forward modelling in the current density
inversion. As for ERT, choices must be made on how data and models are weighted and regularised
during the inversion. In this study, we run unconstrained (no prior information) inversions for all
the time steps with a regularisation (smoothing using the first derivative). The numerical routine
includes a "pareto" functionality wherein regularization and model-to-measurement fit are traded
off to estimate the optimum regularization weight *wr*. The code used for this inversion is available
at https://github.com/Peruz/icsd.
**2.6.3.     Calibration of petrophysical relationships**
In order to estimate the soil water content in the rhizotron during the experiment, we needed to
adopt a suitable constitutive model, starting from the available electrical resistivity measurements.
Archie's (1942) law (eq. 1) is a widely used empirical relationship that relates the electrical
resistivity ($\rho$) of a bulk material to its porosity ($\Phi$), the contained fluid (water) electrical resistivity
($\rho_{fl}$) and the fluid saturation (S). Archie's parameters $a$, $m$, and $n$ are empirically derived, generally
named as follows: $a$ is the tortuosity factor, $m$ is the cementation exponent and $n$ is the saturation
exponent.

$$\rho = a\rho_{fl}\phi^{-m}S^{-n} \tag{1}$$

We calibrated these parameters experimentally, as usually done, by collecting water saturation-ER
values over different soil samples. The sample holder (a cylinder of 150 mm inner height and 41
mm inner diameter) allows for a four-point measurement of the ER converted to apparent electrical
resistivity using the appropriate geometrical factor.  The adopted water electrical conductivity is
known and fixed (594 $\mu$S/cm at ~25°C). Porosity was assumed to be equal to 0.55, which is  the
same of the soil mixturein the rhizotron. The sample was initially saturated to field capacity and
progressively desaturated. The field capacity was estimated by gravimetric method approximately
at 40% of volumetric water content (m$^3$/m$^3$). In total, 6 measurements were collected at respectively





40, 33.6, 29.7, 28.2, 25.2, 22.4% of volumetric water content ($m^3/m^3$). The obtained data are fitted
with a least square optimization (using the Scipy library byVirtanen et al., 2020).  Here we assume
$a$ equal to 1 (consistent with the theoretical value), while the exponents $m$ and $n$ are bounded during
the optimization process to respectively [1.3-2.5] and [1 - 3]. With a coefficient of determination
$R^2$ of 0.97 (figure not shown), we obtained values of 1.9 and 1.2 respectively for $m$ and $n$.

## 3.   Results

### 3.1.   Physiological response

Photographs of the plant at the beginning and at the end of the experiment show the increment of leaf area
extension of the upper partaerial part. The weekly measurements show a linear trend with time of the
estimated total LA ($cm^2$) whichever the model used (Fig. 2). At the end of the experiment water stress
symptoms were were visible on some leaves.
As for the root system, the depth variations could not be precisely assessed during the course of the
experiment. We observed that: (i) roots reached the bottom part of the rhizotron; (ii) spread all over the
rhizotron with a network of primary, secondary, and root hairs without any given architecture (some roots
grew vertically, others in diagonals); (iii) the roots kept a white appearance with apparently no lignification
even for the largest roots (>=3mm).



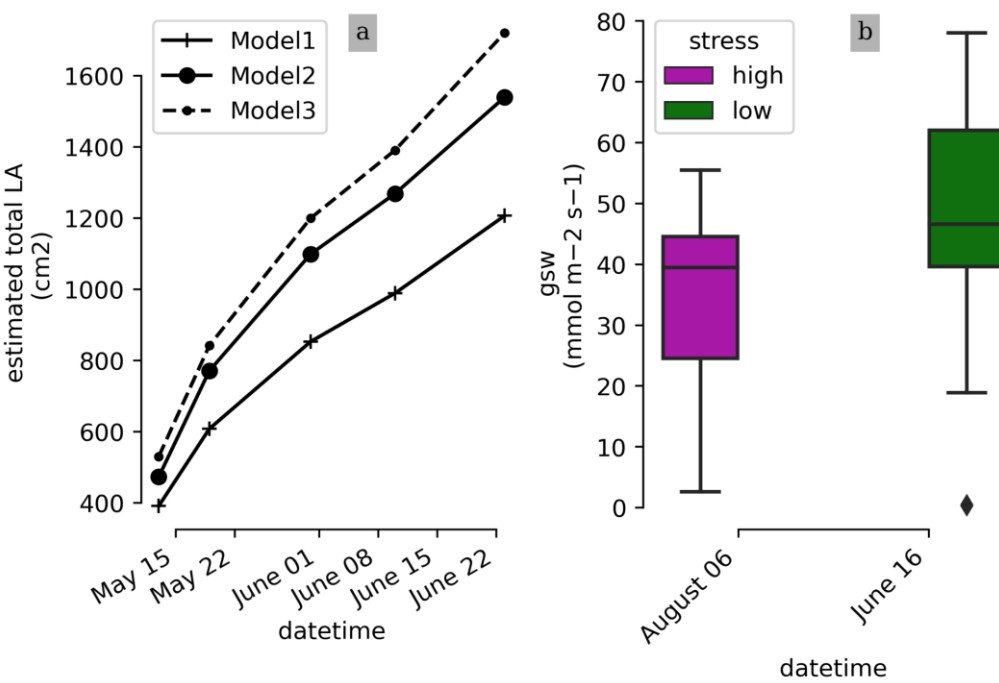

**Figure 2: (a) Time evolution of the estimated total leaf surface area (LA) for three different model estimators. (b) leaf stomatal conductance (High and low stress distributions are significantly different with a T-test p-value = 4.3.10⁻³ )**

The measurements indicate that the plant is under high water stress at the end of the irrigation cycle (one week after the last partial irrigation, on June 8,2022), and under lower water stress one day after irrigation (on June 16, 2022). The mean, min, and max values of the stomatal conductance (*gsw*) values are 37.8; 23.3; 55.5 mmol $m^{-2}$ $s^{-1}$ before irrigation, respectively, and 50.6; 18.9; 78.1 mmol $m^{-2}$ $s^{-1}$ after irrigation, respectively. The result of the T-test shows that their mean values are significantly different (p-value = 4.3.10⁻³).

### 3.2.    Transpiration rate

No pre-processing of the raw data is needed for their interpretation. Fig.3 shows that, on average,  during a PRD cycle (about one week), 0.5 kg of water transpired. Also, the weight data show that the total weight is decreasing from one cycle to the next, as expected, due to the PRD protocol. Although the total water content



is decreasing, the transpiration rate (slope of the weight variations) remains constant for each cycle. At the
very end of the experiment from July 9, an inflexion point is observed and the weight stops decreasing.
Zooming on a shorter time window, the variation of the raw data weight clearly shows day/night patterns
triggered by the hours when the light is switched on/off. On average, the water lost during the day is nearly
20 times more than during the night (0.09 kg/day against 0.005 kg/night). Note that there is no distinction
between the hours of the day (due to artificial lighting).





**Figure 3: Raw scale data collected over the course of the experiment (a) and a zoom on the weekfrom June 20 to 25, where day and night periods are respectively highlighted by the green and red shaded areas. (b) Calculated daily mean transpiration (d_(weight)/dt) during the day (green) and night (orange) periods.**






### 3.3. Time-lapse ERT

In general, the ERT data quality is very good with a small percentage of total measurements exceeding a reciprocal noise level of 5% (see Fig. A1 to A11) and with each inversion resolved within 2/3 iterations. Figure 4shows the trend for the PRD cycles (from cycles -1 to 8) for the mean average electrical conductivity (in mS/m) for both the wet and dry sides of the rhizotron, taken as an average of each half of the ERT inversion mesh elements. When PRD is applied over only two holes (from cycle 3) the irrigated side shows a clear increase in electrical conductivity. To a much lower degree, the dry side is also affected by the water input, likely due to water redistribution during drainage. When available, the temporal dynamics between two irrigations show that the conductivity is decreasing rapidly on the irrigated side during the 2 first consecutive days and more slowly afterwards (cycles C5/6 and C7/8 respectively; Fig. 4c and Fig. 4d). As some water infiltrates also on the dry side, we also observe an increase in conductivity in it. At the end of each cycle (the cycle length is about 7 days), the rhizotron returns to the equilibrium condition, with a more homogeneous and stable average conductivity equal to 30 mS/m (mean of the dry and wet sides). This is generally true for all times, except at the end of the experiment, cycles 7 and 8, when the two sides are in different conditions.

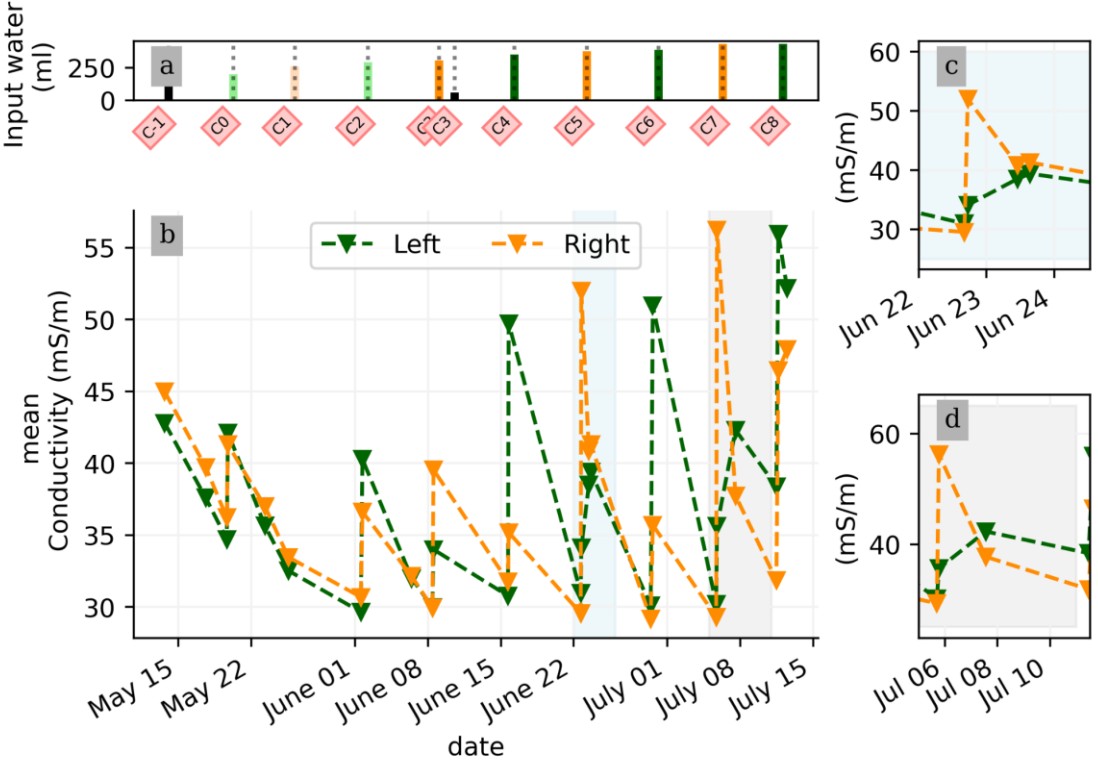

**Figure 4: (a) Evolution of the quantity (in ml) of water input, spatially distributed with alternating between left (green) and right (orange) before and during the PRD irrigation. (b) Evolution of the mean conductivity (mS/m) average on each side, markers show the acquisition time. (c) and (d) are inset zooms showing changes before and just after the irrigation event.**

We selected a time window between 29 June and 5 July showing the spatial variations of the electrical resistivity before and after an irrigation event (Fig. 5). Before the irrigation, the top and lateral boundaries of the rhizotron exhibit higher ER (50 Ohm.m) than the central part (25 Ohm.m). One hour afterwards (+ 1H) the ER of the irrigated side had dropped by 20%.

All time-lapse inversions before/after irrigation are shown in Appendix A, including before the PRD. They all show that a decrease in ER is associated with irrigation patterns while an increase in ER has a more complex spatio-temporal dynamics, not systematically associated with irrigation patterns. Changes in ER after six days (day +6) show that RWU effects are not limited to the irrigated part since the increase of resistivity was also observed on the dry part. Note from a visual inspection of the rhizotron a water table



forms at 0.4 m where the soil is saturated. This saturated zone level is not affected by the irrigation as no

increase after irrigation, and no decrease by the end of the irrigation cycles are visible. We assume that most

of the water fluxes were connected to the unsaturated part.



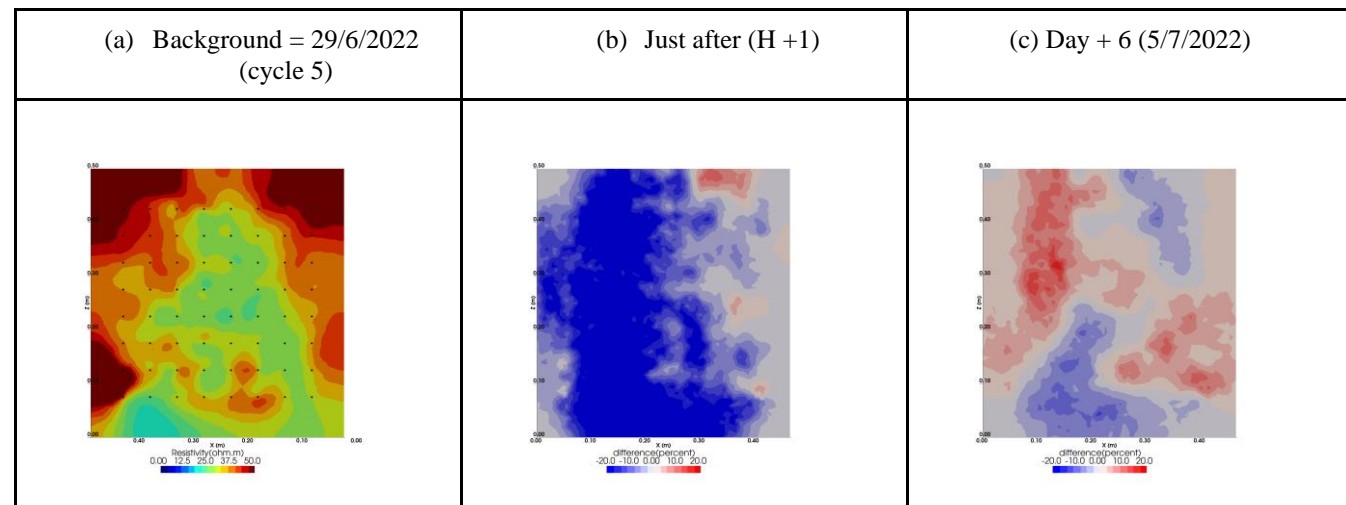


**Figure 5: Spatial distribution of the resistivity (in Ωm) and changes (in %) in electrical resistivity obtained by a time-lapse inversion between cycles 5 and 6 following partial right irrigation of the rhizotron. Time steps correspond to measurements before (a), after one hour (b) and after 6 days (c).**


### 3.4.    Time-lapse ECI


Figure 6shows the trend of the horizontal location (x coordinate) of the centre of mass of current density

during the PRD cycles (from -1 to 8), after the alternative wetting events on the left and right sides of the

rhizotron. The soil CSD is not shown as it is always pinpointed to the location of the injection electrode

whatever the irrigation pattern, as expected (Figure 7abc). This result confirms the quality of the estimated

ER background values used for the ECI forward model. For the stem injection, the centre of mass of the

current source density is distributed equally from left to right except for cycle 3 when most of the current is

located on the left (see Fig. B1 to B4). Conversely to ER variations, the irrigation pattern does not





significantly affect the current density distribution. The same applies to the temporal dynamics between two

irrigations where the current density centre of mass is stable and distributed equally on both sides, as shown

in Fig. 7. All the time-lapse inversion results of current density for the soil and the stem injection are shown

in Appendix B.

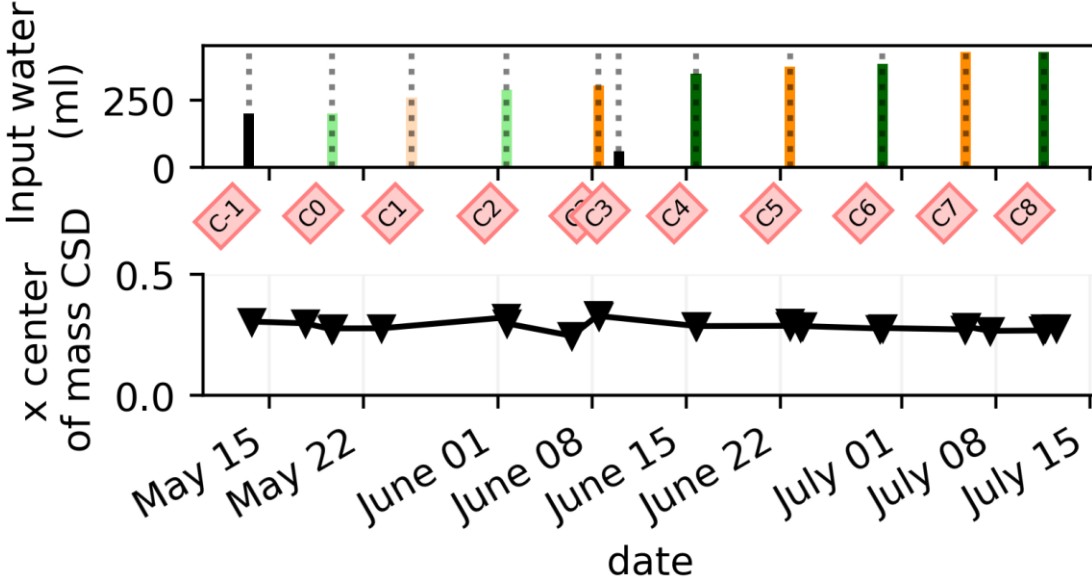

**Figure 6: (a) Evolution of the quantity (in mL) of water input spatially distributed alternatively between left (green) and right (orange) during the PRD irrigation. (b) Evolution of the centre of mass (in the x direction) of the current density, while cross markers show the acquisition times. Cycle 5 and 6 windows were selected for the MALM time-lapse spatial analysis (Figure 7).**



| Background = 29/6/2022 (cycle 5) | Just after irrigation (H+1) = 29/6/2022 14h15 | Day + 6 = 5/7/2022 16h35 |
|---|---|---|
| a (soil control, 10h24) | b (soil control, 15h02) | c (soil control, 17h55) |
| | | |
| d (stem, 10h14) | e (stem, 14h50) | f (stem, 17h15) |
| | | |

**Figure 7: Spatial distribution of the CSD between cycles 5 and 6 following partial (right) irrigation of the rhizotron for the soil control injection (a,b,c) and the stem injection (d,e,f). The larger spread of current sources in the stem injection (d, e, f) compared to soil control injection (a, b, c), demonstrates that the root system plays a key role in the distribution of the current source in the soil. Time steps correspond to measurement before (a,d) irrigation, one hour after irrigation (b,e), and after 6 days (c,f). The regularisation parameter *wr* is fixed to 10 for both cases (see section 2.6.2 for the choice of *wr*).**

### 3.5. Correlations between soil parameters and estimated transpiration rates.

This section aims at drawing correlations between the soil parameters (ER, SWC, and CSD) and the transpiration estimated from the rhizotron weight data. We do not account for the weight variations due to the plant and root growth material (as this can be considered negligible relative to water dynamics).



Figure 8 shows the relationship between the variation between two consecutive measurements of the weights

with the variations of average electrical resistivity (Fig.8a, $R^2$=0.76, p-value=6.5 x $10^{-5}$) and those of

resistivity-derived average water content (from Archie's law - Fig.8b, $R^2$=0.815, p-value=6.8 x $10^{-6}$). An

increase in weight over time is positively correlated with an increase in in resistivity and water content

meaning that the changes in resistivity are mainly associated with transpiration (rather than changes in soil

structure or other parameters).

For each node of the mesh, ER values are translated to SWC using Archie's law with the calibrated

parameters *m* and *n* (see Sect. 2.6.3). To simplify, we assume that both porosity and fluid water conductivity

are homogeneous in space and time (i.e no mixing between the tap water used for cycle 3 and the nutrient

solution for all the other times). The maximum SWC observed after irrigation is about 0.42 $m^3/m^3$ (figure

not shown). The minimum SWC of about 0.25 $m^3/m^3$ is repeatedly observed (see Fig. C1) just before each

irrigation, meaning that the driest times are below field capacity conditions (estimated at 0.4 $m^3/m^3$).

Translated ER to SWC improve slightly the strength of the correlation with the transpiration (variations of

weight) due the non-linear nature of Archie's law.




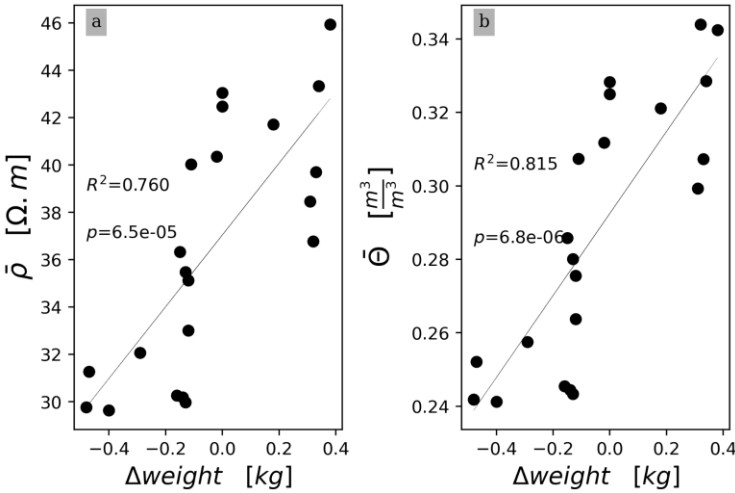






**Figure 8: Relationship between time variation of weight and the time variation of the average electrical resistivity (a) and of the average estimated water content (b) in the rhizotron. Straight lines show the linear regression fit obtained. All cycles are considered.**

Figure 9 shows the relationship between the variation of the percentage of the current sources carrying at least 1% of the total density ($Ns_1$) used as an estimator for current density dispersion with respect to the SWC. For the soil injection (blue dots), $Ns_1$ is relatively constant between 5 to 10% of the total number of possible injection nodes (grey area) irrespective of the SWC values (spanning the whole range of volumetric water content from 0.25 to 0.42). For the stem injections, we distinguish between values after (black triangles) and before (grey triangles) cycle 3, for which no stress has been applied (grey triangle Fig. 9). For the stem injection, for cycles where stressed was applied, $Ns_1$ is 4 to 5 times (appr. 25 to 30% of the sources carrying at least 1% of the total current density) more than for the soil. For cycles where stressed was not applied (i.e. < cycle 3), for the stem injection, $Ns_1$ is distributed between 5 and 25%. No trend between the current spread with increasing water content levels is visible. From Figure 9, we noticed that the current spreads less before the actual PRD (grey triangles) started than after (black triangles).

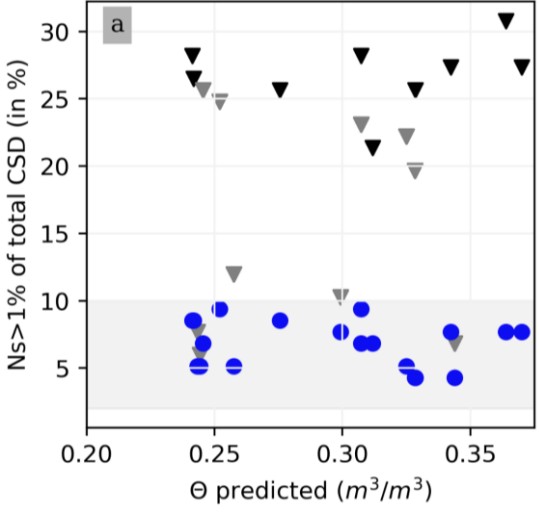

**Figure 9: (a) Relationship between the number of the current sources (Ns) carrying at least 1% of the total density (A.m$^{-2}$) with respect to the estimated SWC (m$^3$/m$^3$). CSD results are obtained after inversion with a regularisation parameter *wr* of 10. Cases of the stem before cycle 3 (grey), after cycle 3 (black) and the soil (blue) injections. All cycles are considered.**





## 4. Discussion

### 4.1. Validity of ERT and ECI in demonstrating the effects of the PRD irrigation scheme

Our first assumption was that the variations in ER (or in SWC inferred from the ER) are relevant as a proxy of root activity. Its validity has been checked against direct observation using the variations of weights measured from the scale data used as an indicator of plant transpiration. On average, in our experiment, the plant maintained high rates of transpiration to about 6 mm/day for each cycle except for the last cycle (number 9) where a decline was observed (Fig. 3). This range is in line with another rhizotron experiment where narrow-leaf lupin plants were grown: Garrigues et al. (2006) measured a mean rate of 3 mm/day. It is commonly found in the scientific literature that changes in ER are associated with root activity (e.g., Michot et al.,2003; Garré et al.,2011; Cassiani et al., 2015; Whalley et al., 2017). Here we had further confirmation of this, with a significant correlation between ER changes and gravimetric soil moisture changes (derived from the load cell) (Fig. 8). The leaf stomatal conductance and visual observation of plant above- and below-ground material growth were additional ancillary data to interpret the general state of the plant. Our observation is in line with the literature i.e. in general, low soil water content (SWC) can lead to drought stress in plants, which can result in decreased leaf stomatal conductance and less transpiration, and vice-versa.

A second assumption was that, when applying the PRD, only one part of the root system would be active and the current injected in the stem would only spread to the side where the root system is irrigated. This assumption was not directly supported by the observations. Figures 6 and 7 show that the influence of the irrigation pattern was negligible on the spatial distribution of the inverted CSD and that the current distribution was not correlated with ER variations. It is true that active roots have higher hydraulic conductivity but on the other hand, increased membrane permeability may encourages current leakage into the soil. We nevertheless noticed that the CSD spatial distribution, while the rhizotron is irrigated at its full





length (cycles -1 to 2), was significantly different from the side irrigation cycles (Fig. B4). Indeed,
homogeneous irrigation without applying stress to the plant results in a very shallow current leakage. This
is a hint that the hydraulically stressed plant tends to have a wider and deeper active root system, even not
necessarily active only on the side where the PRD is temporarily applied. Possibly the reaction of the plant
to the changing side is too slow to show up in our measurements, but the reaction to general stress is apparent.

**4.2.     Effect of soil water content**
Soil water content can affect the distribution of the current leakage by influencing the minimum resistance
pathways, i.e., whether roots and/or soil provide the minimum resistance to the current flow. Literature
reports that electrical capacitance method better estimates crop root traits under dry conditions (Gu et al.,
2021). In order to make a comparison with capacitance studies, we assumed that if the current distribution
remains unchanged (i.e. leaking into the same areas), there must be minimal changes in the electrical
capacitance. In this study, supposing no impact of the initial model, Fig. 9 shows that there is no apparent
effect of the soil water content on the current density distribution. Note that the soil water content estimated
is the bulk contribution of roots and soil, as only one pedophysical relationship was used, while recent studies
tend to show that mixed soil-root pedophysical relationships are preferable (e.g. Rao et al., 2018). This is
clearly limiting our ability to interpret the independent contribution of the soil and the roots, yet this does
not limit our ability to identify zones where water availability leads to root water uptake.
**4.3.     Possible mitigation of the PRD effect**
In general, a PRD irrigation experiment must comply with two criteria: (1) a minimum soil water content to
trigger a physiological response and, (2) a distinction between a wet and a dry side (Stoll, 2000). While the
first criterion complied in our experiment, the second did not. And the latter is a very interesting piece of
evidence. The following considerations apply.





(1) According to McAdam et al. (2016) and Collins et al. (2009), ABA is triggered even by mild soil stress values. Consequently, plants adapt the hydraulic conductivity of their roots as well as that of the soil in their vicinity through exudates (Carminati and Javaux, 2020). Results from previous irrigation experiments using PRD have shown that changes in stomatal conductance and shoot growth are some of the major components affected (Düring et al., 1996). In our experiment, the shoot growth fitted with the conventional leaf area and growth models, except at the end of the experiment when signs of water stress were visible on some leaves. The magnitude of the shoot growth is correlated with the number of roots. Drought may cause more inhibition of shoot growth than of root growth (Sharp and Davies, 1989). Although the root system was already well developed it is not possible to exclude its development as a factor influencing the CSD distribution.

(2) The spatiotemporal analysis of the ER showed that the water changes were not limited to root effects. Water redistribution from dry to wet in the soil and from shoot to dry roots (Smart et al., 2005, Lovisolo et al., 2016) may have occurred (Fig. A1 to A11). Additionally, capillary rise may have taken place due to the presence of a saturated zone at the bottom of the rhizotron. Due to the fact that water drained on both sides, RWU was not only vertically distributed but also horizontally. The range of water content varied significantly with a minimum SWC of about 0.25 $m^3/m^3$, repeatedly observed just before each irrigation meaning that the driest times are below field capacity conditions (estimated at 0.4 $m^3/m^3$). Drying half of the root system resulted in a reduction of the stomatal conductance (based on the mean of the distribution) of the order 5 mmol $m^{-2}s^{-1}$ after a 1 week cycle. Given the stress applied, the ER changes highlighted that root played a major role in the wine plant survival and evidenced strategies of adaptation. Indeed, the plant was able to change its water uptake zones depending on the water availability, from all places, not only from the alternate irrigated areas.

(3) Finally, in order to know if the PRD conditions are met it would have been important not to neglect the different states of root growth, and root renewal (because of renewal and decay) with respect to





the geophysical data. Nevertheless, this would have required opening and scanning the rhizotron

with conventional methods. Finally, we did not make a distinction between the hours of the day

although the changes observed for the irrigation are rapid, usually at the hourly scale, and could be

similar for RWU.

**4.4.**        **Performance of the acquisition protocol and the processing**

We discuss here how the quality of the recovered current density models by evaluating the performance of

the protocol and the processing. First, it is important to note that although the ERT data quality was really

good (very few reciprocals were rejected, see Table A1), the inverted model was not perfect and this

ultimately has an impact also on the ECI forward model. The algorithm has been tested already in a rhizotron

experiment and is capable of distinguishing between punctual sources with the lowest current carried of 5%

of the total current (Peruzzo et al., 2020). The CSD resolution, of course, matches the electrode interspace

(in this case 5cm) and the smoothness constraint does not impact the simulation of point source

reconstruction. We adopted an inversion without any prior information to recover the current density. Only

model smoothing was applied by weighting the model data by an optimal factor of 10 inferred from an L-

curve analysis. Similar to the ERT inversion, the ICSD the problem is also ill-posed. In this case, the 4-

electrodes setup ensures that the current will flow through the plant after injection, regardless of the contact

resistance. However, the accuracy of the measured data may be impacted by contact resistance, as errors in

the measured resistance will negatively affect the quality of ERT and ICSD inversions. The impact is more

pronounced on ICSD, as it is dependent on ERT. Lastly, because the box is relatively small and no-current-

flow boundary conditions (Neumann) are imposed, we may expect an effect due to the position of the return

electrode where the current is attracted due to the strongest gradient nearby (Mary et al., 2019b).





**4.5.  Outlook**

In order to strictly correlate PRD effects with geophysical measurements, one should consider a physical barrier to separate the two sides of the rhizotron to a split-roots configuration. Another option is to increase the lateral size to prevent redistribution or to use a very percolating material such as glass beads, gravels or coarse sands. This should be carefully considered, as the rhizotron must also be an environment where plant growth is possible under "natural" conditions, and for this some water retention capacity is needed for the soil. A larger drainage capacity would simplify the interpretation as no-water redistribution from one side to the other can occur. Although considering a barrier is technically possible, it would require a more complex inversion scheme of the ERT and ECI considering that no electrical current can flow from side to side. One could also consider increasing the measurement frequency to catch processes at an hourly scale and comparing day/night measurements, particularly those associated with water redistribution from the stem back to the roots at night when transpiration is reduced and its effect on the water status of the roots. As we have seen that most of the water changes occurred in the day consecutive to the irrigation, catching rapid changes of ER would help drive a conclusion on how much ECI is connected to the active root zone. Finally, in order to draw robust statistical conclusions, the experiments should be replicated for multiple plant samples.

**5. Conclusion**

The study aimed at understanding the current path in the root system and active root zones using geoelectrical imaging, considering soil water content and irrigation regimes. Electrical Resistivity Tomography (ERT) is sensitive to both irrigation and RWU processes. The ECI model uses a physical approach to measure current density after stem stimulation. The CSD was very different from the control soil injection to the stem injection but nevertheless did not correlate with PRD cycles as originally expected. We demonstrate that under mild stress conditions, it is practically impossible to spatially distinguish the PRD effects using the ECI. We only evidenced that the Current Source leakage depth varied during the course of the experiment but without any significant relationship to the Soil Water Content changes or evaporative demand. A few aspects of the





experiment would gain to be more closely studied such as the water redistribution that possibly also affects current distribution.
In the future, we expect to improve our understanding by coupling the geophysical experiment with an unsaturated soil-plant-
atmosphere model.

## 6. Appendices

**Appendix A: Time-lapse ERT inversion results**

As we selected only one cycle in the manuscript, we report here further details about the time-lapse ERT inversion results for
all the cycles. The inversion procedure is equivalent to the one described in Sect. 2.6.1 of the manuscript (Data processing -
Analysis of the ERT data). All time-lapse inversion models are plotted with a unique scale ranging from -20 to 20% of changes.

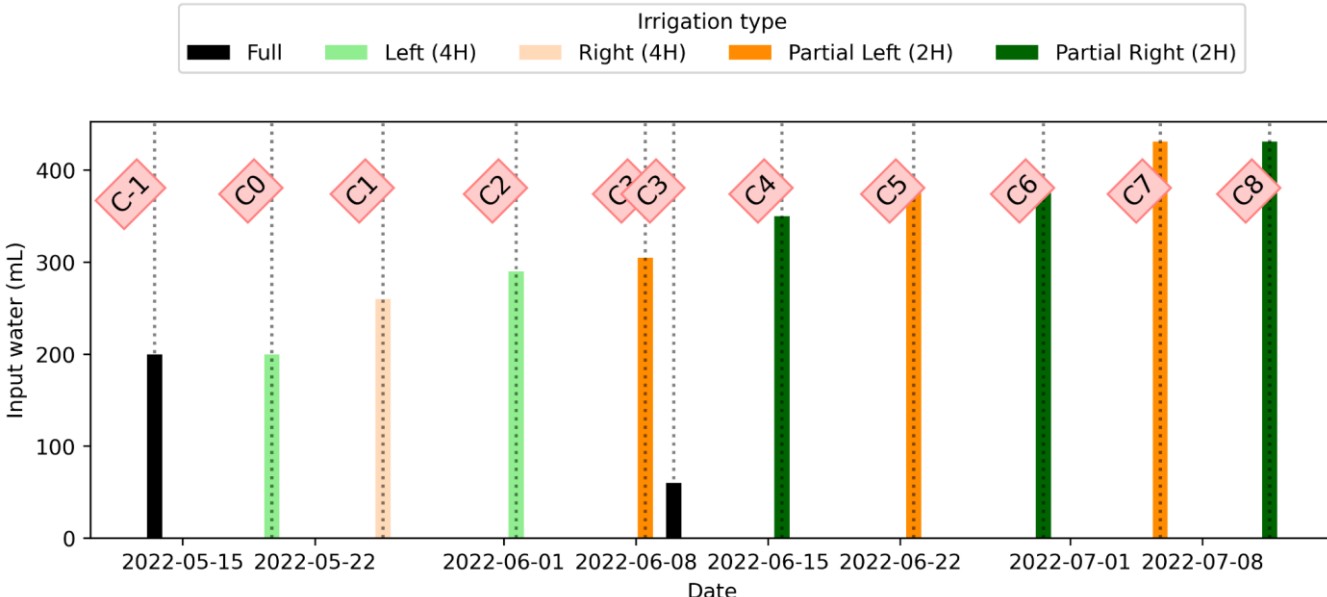


**Figure A1: Evolution of the quantity (in mL) of water input spatially distributed with an alternate between left (green) and right**
**(orange) during the PRD irrigation. The black bars hold for full-width irrigation (over all the holes, see fig. 1 manuscript), light**
**green and orange bars hold for irrigation over the 4 sides of holes, and dark green/orange for 2 holes irrigation.**



| Background = 13/5/2022 16h25 (cycle -1) | Day + 4: 17/05/2022 15h00 (cycle -1) | Day + 6: 19/5/2022 15h38 (cycle -1) |
| --- | --- | --- |





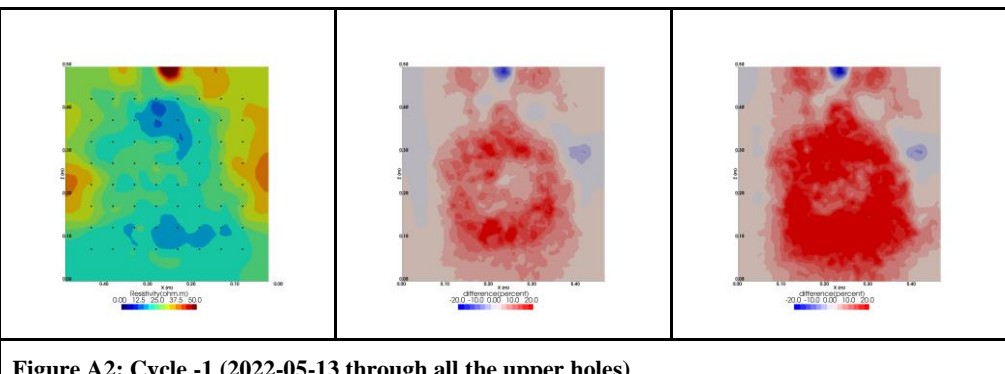

**Figure A2: Cycle -1 (2022-05-13 through all the upper holes)**



| Background = 19/5/2022 15h38 (cycle -1) | 19/5/2022 18h20 (cycle 0) | 23/5/2022 07h45 (cycle 0) | 25/5/2022 13h30 (cycle 0) |
|---|---|---|---|
| | | | |

**Figure A3: Cycle -1 to 0 (partial irrigation: 19/05/2022 17:00-17:30 200 ml through the first 4 upper holes (left side), no outflow through 72)**








| Background = 25/5/2022 13h30 (cycle 0) | Day + 5: 1/6/2022 12h50 |
|---|---|
| | |

**Figure A4: Cycle 0 to 1 (partial irrigation: 25/05/2022 14:30-14:15 260 ml through the last 4 upper holes (right side), no outflow through 72)**


| Background = 1/6/2022 12h50 (cycle 1) | H + 4: 1/6/2022 16h35 (cycle 2) | Day + 5: 6/6/2022 10h15 (cycle 2) | Day + 7: 8/6/2022 10h00 (cycle 2) |
|---|---|---|---|
| | | | |

**Figure A5: Cycle 1 to 2 (partial irrigation: 01/06/2022 15:50-16:10 290 ml through the first 4 upper holes (left side), no outflow through 72)**










| Background = 8/6/2022, 10h00 (cycle 2) | H+2: 8/6/2022 12h30 (cycle 3) |
|---|---|
| 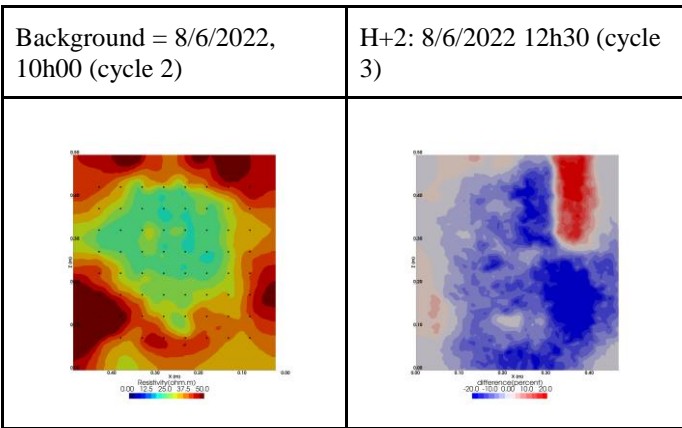 | |

**Figure A6: Cycle 2 to 3 (partial irrigation: 08/06/2022 11:50-12:00 305 ml through the last 2 upper holes (right side))**


| Background = 15/6/2022 16h20 (cycle 3) | H +1: 15/6/2022 17h50 (cycle 4) | 22/6/2022 16h10 (cycle 4) |
|---|---|---|
| | | |

**Figure A7: Cycle 3 to 4 (partial irrigation: 15/06/2022 17:25-17:45 350 ml through the first 2 upper holes (left side))**















| Background = 22/6/2022, 16h10 (cycle 4) | Just after (H+1 i.e 17h30) | 23/6/2022 (10h55, Day + 1) | 23/6/2022 (15h20, Day + 1) | 29/6/2022 (9h30, Day + 7) |
|---|---|---|---|---|
| | | | | |

**Figure A8: Cycles 4 and 5  time-lapse inversion (partial right side irrigation)**

| Background = 29/6/2022 (cycle 5) | Just after (H +1) | Day + (5/7/2022) |
|---|---|---|
| | | |

**Figure A9: Cycles 5 and 6 time-lapse inversion (partial left side irrigation)**





| Background = 5/7/2022 (cycle 6) | Just after (H +1) | 07/07/2022 (Day + 2) | 11/7/2022 (Day + 6) |
| --- | --- | --- | --- |

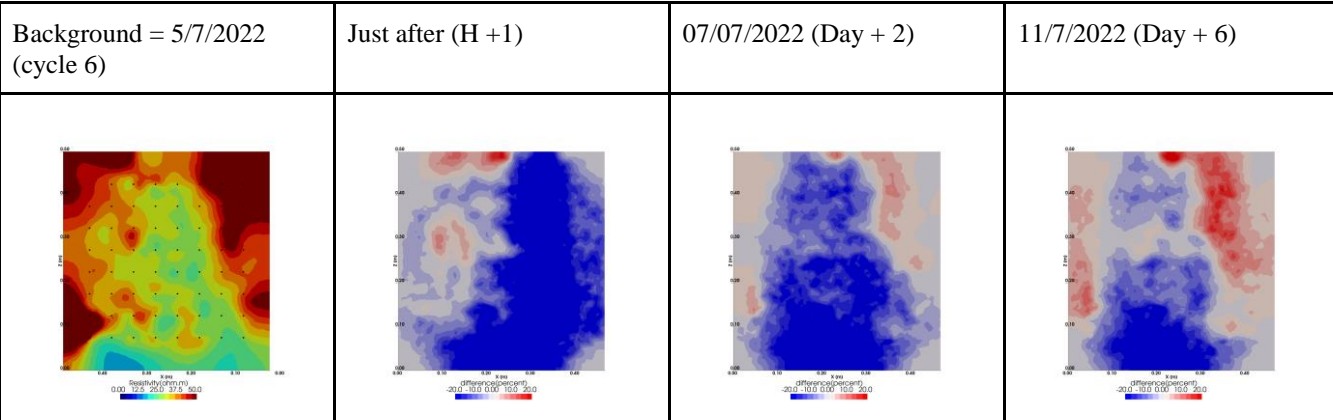

**Figure A10: Cycles 6 and 7 time-lapse inversion (partial right side irrigation)**



| Background = 11/7/2022 (cycle 7) | Just after (H +1) | 12/7/2022 (Day + 1) |
| --- | --- | --- |

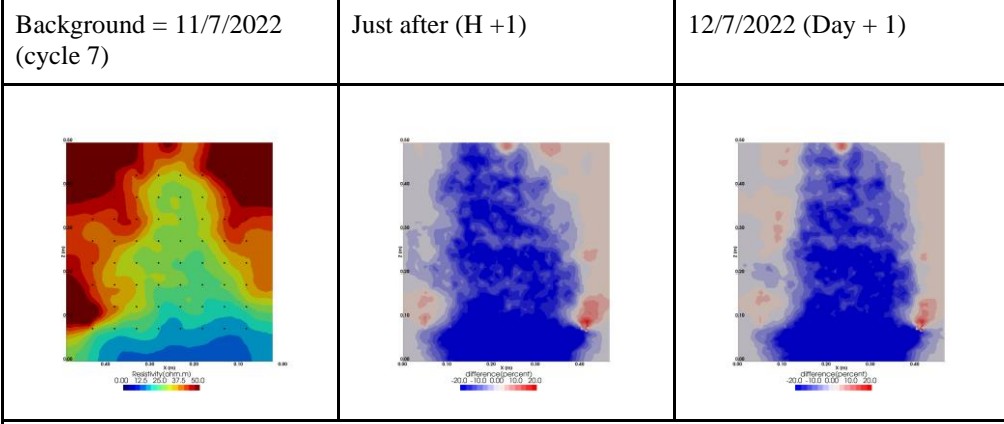

**Figure A11: Cycles 7 and 8 time-lapse inversion (partial right side irrigation)**












| Date | RMS (%) | # measurements read (over 2484) |
|---|---|---|
| 2022-06-01 12:50:00 | 1.36 | 2048 |
| 2022-06-01 16:35:00 | 1.15 | 1920 |
| 2022-06-06 10:15:00 | 1.53 | 2268 |
| 2022-06-08 10:00:00 | 1.41 | 2230 |
| 2022-06-08 12:30:00 | 1.16 | 2028 |
| 2022-06-15 16:20:00 | 1.08 | 2137 |
| 2022-06-15 17:50:00 | 1.47 | 1493 |
| 2022-06-22 16:10:00 | 1.38 | 2109 |
| 2022-06-22 17:21:00 | 1.14 | 1372 |
| 2022-06-23 10:55:00 | 1.48 | 2229 |
| 2022-06-23 15:20:00 | 1.38 | 2268 |
| 2022-06-29 09:30:00 | 1.27 | 2075 |
| 2022-06-29 14:15:00 | 2.04 | 2027 |
| 2022-07-05 16:35:00 | 1.7 | 2067 |
| 2022-07-05 18:25:00 | 1.85 | 980 |
| 2022-07-07 13:15:00 | 1.98 | 2225 |
| 2022-07-11 11:20:00 | 2.5 | 2093 |
| 2022-07-11 15:50:00 | 2.72 | 2238 |
| 2022-07-12 12:00:00 | 2.68 | 2255 |

**Table A1: Table summarising the final RMS and the number of data used for each individual inversion**





**Appendix B: Inversion of current density (ICSD)**



As we selected only one cycle in the manuscript, we report here further details about the time-lapse ICSD inversion results for
all the cycles. The inversion procedure is equivalent to the one described in Sect. 2.6.2 of the manuscript (Data processing -
Analysis of current density) and we invite the reader to refer to Peruzzo et al. (2020) for a full description of the algorithm.
Furthermore, we extend the analysis showing the effect of the model regularisation (smoothing). Figures B1 and B2 show the
current density evolution with the time respectively for the stem and the soil injection with a regularisation parameter of 1.
The same is for Figures B3 and B4 with a regularisation of 10.





**Figure B1: variations of the CSD for all the time steps (all cycles) during the stem injection. Inversion is unconstrained; data-model weighting factor (wr) is set to 1.**





**Figure B2: variations of the CSD for all the time steps (all cycles) during the soil control injection. Inversion is unconstrained; data-model weighting factor (wr) is set to 1.**












**Figure B3: variations of the CSD for all the time steps (all cycles) during the soil control injection. Inversion is unconstrained; data-model weighting factor (wr) is set to 10.**



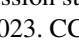



**Figure B4: variations of the CSD for all the time steps (all cycles) during the stem injection. Inversion is unconstrained; data-model weighting factor (wr) is set to 10.**











**Figure B5: Evaluation of the quality of the CSD inversion for the acquisition date 2022-07-11. The linear correlation coefficient is always > 0.95 for all the time steps.**





**Appendix C: Soil Water Content converted variations**





**Figure C1: (a) Evolution of the quantity (in mL) of water input spatially distributed with an alternate between left (green) and right (orange) during the PRD irrigation. The black bars hold for full-width irrigation (over all the holes, see fig. 1 manuscript), light green and orange bars hold for irrigation over the 4 sides of holes, and dark green/orange for 2 holes irrigation. (b) Evolution of the mean SWC (m3/m3) average on each side, markers show the acquisition time.**

**7. Data availability**
Codes and data to reproduce figures articles are available in the Zenodo data repository (link to come after decision).





*Competing interests*

The authors declare that they have no conflict of interest.

*Author contribution*

BM, VI, LP, FM, BR, CC, YW and GB designed the experiments, and BM, VI, BR and FM carried them out. BM, LP, GB , CC developed the model code and performed the simulations. BM prepared the manuscript with contributions from all co-authors for writing – review & editing.

*Acknowledgments*

Benjamin Mary acknowledges the financial support from European Union's Horizon 2020 research and innovation programme under a Marie Sklodowska-Curie grant agreement (grant no. 842922).

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
