# Peer review of "Imaging of the electrical activity in the root zone under limited water"

_Biogeosciences, 2023_

## Author Comment (AC1)

In this paper the authors present a very nice laboratory-based experimental study to assess the utility of novel electrical methods for monitoring and quantifying root water uptake and plant stress under partial root zone drying and irrigation schemes in an effort to study plant phyisology. They use both electrical resistivity tomography and electrical current imaging, two geophysical techniques that through a computational inversion provide visual and quantifiable evidence for the distribution of changes in electrical resistivity as a proxy for water content and current source density as a proxy for active root pathways, respectively. This experiment was conducted using a single vine grown in a small rhizotron over the course of several months in 2022. In addition to repeated cycles of irrigation and geophysical data collections, the transpiration was monitored using the weight of the rhizotron. Stomatal conductance and leaf area were also measured during one cycle to capture plant stress during a partial root drying event. Aside from a few grammatical errors and some issues with figure quality, this paper is an important addition to the biogeosciences literature, in particular for scientists taking advantage of novel geophysical techniques for noninvasive methods.

We thank the reviewer for her words of appreciation and for her extensive review. We have carefully considered and addressed all of the reviewer's comments, which we believe have significantly enhanced the quality of the manuscript.

The main changes we have made include:

(i) improvement of figures' quality (consistent colours), proofreading the text for typos and rephrasing it when necessary.

(ii) Modifying the description and discussion of our experiment by replacing "PRD" with "root water limited availability," as indicated in the revised title. This change had only a minor impact on the actual results, as they were already aligned.

(iii) Responding to the reviewer's comment by incorporating new references in the introduction that briefly cover various methods used to measure root physiology, anatomy, and biomass. By discussing the assumptions associated with these methods, we are able to establish up-to-date parallels between current pathways and water pathways.

(iv) Completely reshaping figures 8 and 9 and discussing the observed current density in this experiment, considering it as a result of both higher transpirational demand and/or drier soil conditions.

These revisions and improvements have enhanced the manuscript, and we sincerely appreciate the reviewer's insightful comments that have contributed to its overall quality.

Major comments:

There are some major inconsistencies and errors in the labelling of 'left' and 'right' irrigation/PRD

that persists in most Tables and Figures which makes it difficult to understand the results. In addition to correcting these issues, interpretation would be easier if the same vertical blue arrows used in Figure 1 was added to the top axis of all the subsequent cross section figures (i.e. Fig. 5, 7, A2-A11). Some examples of the inconsistencies:
Numbering of irrigation cycles:
In Fig 1, 'Cycle 1' is depicted as left-side irrigation, yet in Table 1 the cycle numbering begins at 0, so that odd numbered cycles are actually right-side irrigation. I would make the initial wetting through all Holes Cycle 0 and then the first left-side irrigation on 2022-05-19 Cycle 1 etc.

Thanks for spotting this error. We corrected the figures and table accordingly.

Color coding of left vs right sided irrigations:
In Figs 4a and 6a, dark green is 'left' and orange is 'right'
In Fig A1 there is a legend at the top indicating the opposite of the caption - that dark orange is partial left and dark green is partial right.

Colors are now consistent between left and right:
- Left = Green
- Right = Orange

| Date (YYYY-mm-dd HH:MM) | Hole (H) location (c.f. Fig. 1) | Quantity (mL)* | Cycle nb |
|---|---|---|---|
| 2022-05-13  16:25 | All |  | 0 |
| 2022-05-19 17:00 | H1;H2;H3;H4 | 200 | 1 |
| 2022-05-25 14:30 | H5;H6;H7;H8 | 260 | 2 |
| 2022-06-01 15:50 | H1;H2;H3;H4 | 290 | 3 |
| 2022-06-08 11:50 | H7;H8 | 305 | 4 |
| *2022-06-10* | All | 60 | (5) |
| 2022-06-15 17:25 | H1;H2 | 350 | 6 |
| 2022-06-22 16:45 | H7;H8 | 375 | 7 |
| 2022-06-29 13:45 | H1;H2 | 386 | 8 |
| 2022-07-05 18:10 | H7;H8 | 431 | 9 |
| 2022-07-11 13:15 | H1;H2 | 431 | 10 |
| 2022-07-12 16:00 | H1-H8 | 200 | - |

[Figure]

Fig A1 was corrected accordingly.

In Table 1, while labelled correctly, it is also color coded the opposite to Figs 4 and 6, so that 'green is right' and 'orange is left'.

Table 1 colors were corrected accordingly. See the previous answer.

Erroneous Figure captions:
Table 1 cycle 2 Date column should be 2022-06-01 instead of 2022-05-01

Well spotted, thanks. Table 1 cycle 2 Date column was corrected accordingly.

Fig 5 and Fig A9 are the same - labelled time lapse between cycles 5 and 6 - however Figure 5 is captioned 'following partial right irrigation' while Figure A9 is captioned 'partial left-side irrigation'.

Fig 5 legend rephrased 'following partial left irrigation'. Note that this did not affect the core text description/interpretation.

The date/time formatting on the header row of the time lapse ERT figures (5, A2-A11) are inconsistent and I find the cycle number labels particularly confusing in this context because the 'background' is labelled as the end of the prior cycle and I don't intuitively think of background is the end of the old cycle. I thought 'between cycles 5 and 6' meant the start of 5 to just before the start of 6, as opposed to the very end of 5 to the very end of 6 which I believe is the intention.

Datetime are now consistent everywhere using the format: YYYY-MM-DD HH:MM, except for the figures 2,3,4,6 and 8 where we used Month-DD format.

Figures A2-A7 label the second image as the next cycle but A8-11 do not and that makes it seem like the cycle number is incorrectly labelled for that entire set of images. Since the time is shown in at least some of the timelapse figures, it would be helpful to have the time of the irrigation in Table 1 (as it is in Table A1) (and to use the same date formatting throughout all figures/tables). I think a combination of the headers would be best i.e. for Fig 5: Background (-1h) = 2022-06-29 16h20, Just After Irrig. (+1h) = 2022-06-29 17h20, Six days after Irrig. = 2022-07-05 17h20, and simple caption this is cycle 6 and add a title to the whole figure that says Cycle 6.

All the time-lapse ERT figures headers were renamed according to the reviewer's suggestion.

| (a) | (b) | (c) |
|---|---|---|
| Background (-4h) = 2022-06-29 9:30 | Just After Irrig. (+0h15) = 2022-06-29 14:15 | 6 days after Irrig = 2022-07-05 16:35 |

[Figure]

Figure 5: Spatial distribution of the resistivity (in Ωm) and changes (in %) in ER obtained by a time-lapse inversion for cycle 8 following partial left irrigation of the rhizotron (2022-06-29 13:45-14:00, 386 ml). Time steps correspond to measurements before (a), -4h min (b) just after irrigation (+0h15) (c) and 6 days after irrigation started.

The abstract is too technical and abbreviated in my opinion. It will probably be unclear to general readers why either ERT or ECI could be useful for understanding root water dynamics so I would advise adding a sentence that introduces the concept of Archie's law.

*We added a sentence to introduce the concept of Archie's law: "To estimate soil water content in the rhizotron during the experiment, we incorporated Archie's law as a constitutive model"*

Further, the term 'current source leakage' is only used twice in the paper, once in the abstract and once in the conclusion although much of the paper focuses on current source density/current density. Making the usage of the electrical methods terminology more consistent throughout the paper would be helpful to the reader.

*The reviewer is right. All "current source leakage" occurrences are now replaced by "current source density"*

Similar to the abstract, the title has an emphasis on specifically imaging 'the active root current pathway' but after reading the paper my takeaway was that the focus of the paper is more of an assessment of electrical methods for assessing root water uptake and observing the patterns of the PRD. To justify this in the title I think the introduction would need more background on the signifcance and meaning of the active root current pathway.

*The reviewer is right. The title is rephrased considering also Reviewer 2 comments.*
*"Imaging of the electrical activity in the root zone under limited water availability stress: A laboratory study for Vitis vinifera."*

Minor comments:

Fig 1: the horizontal flux arrows are a bit confusing since I would expect at least some of the flux coming from the surface to be vertical. I would consider simply removing those arrows or replacing them with something more realistic.

The figure was improved accordingly.

Fig 5 (and A2-A11): the axes labels and legend are too small and low resolution to read clearly

The reviewer is right. We improve the axis label and color bar quality.

[Figure]

| (a) | (b) | (c) |

| Background (-4h) = 2022-06-29 9:30 | Just After Irrig. (+0h15) = 2022-06-29 14:15 | 6 days after Irrig = 2022-07-05 16:35 |
|---|---|---|

[Figure]

Ln 20: Based on the other figures/tables it seems like there are only 4 or 5 PRD cycle pairs, not six as stated here

The sentence has been rephrased to make this clearer:
"*In controlled laboratory conditions, using a rhizotron built for geoelectrical tomography imaging, we monitored the spatio-temporal changes in soil electrical resistivity (ER) for more than a month corresponding to 8 alternating water input cycles.*"

Ln 23: 'Current Source Leakage Depth' is a very technical term that is not going to be understood by most readers without some explanation. I would consider rephrasing or adding some clarification.

The reviewer is right. We replaced '*Current Source Leakage Depth*' with '*Current Source Density spatial distribution*'

Ln 88: The concept of 'active roots' is very important for the paper but there is not really a definition of what makes roots 'active' vs inactive. The sentence on this line explains how water moves in active roots but doesn't make it clear whether inactive roots are those that will never take up water, or those that aren't taking up water at the moment.

The reviewer is right, the concept of active roots needs to be introduced further. We added 2 sentences to make it clearer:

"*The concept of active roots has been previously employed by several authors (Frensch and Steudle, 1989; Doussan et al., 1998; Garrigues et al., 2006; Srayeddin and Doussan, 2009) to characterize the spatial variability of root water uptake. In this context, plants adapt by reducing*"

*radial conductivity in dry regions, enabling them to redirect their uptake towards wetter areas with higher soil conductivity. This mechanism allows plants to maintain a consistent rate of water uptake while sustaining higher plant water potentials."*

*"Fine root connections and mycorrhiza facilitate the efficient transfer of injected current into the soil at contact points between roots and the soil, resulting in a distribution of current sources within the ground."*

Lns 109-113: consider rephrasing this sentence to make it more clear

Sentence rephrased:

*BEFORE: "Since then, contrasted experimental results opposed on the relationship between root capacitance ("ECroot") and root traits in various crops, particularly because of studies supporting the major contribution of the stem compared to the roots on the total ECroot measured and the possible current leakage at the proximal part"*

*AFTER: "Contrasting experimental results have challenged the relationship between root capacitance ("ECroot") and root traits in different crops, with studies highlighting the potential contribution of the stem, rather than the roots, to the overall measured ECroot and the occurrence of current leakage at the proximal part (Urban et al., 2011; Dietrich et al., 2018; Peruzzo et al., 2020)."*

Ln 122: this is the first use of term 'electrical current leakage' which has not yet been explained, particularly as it relates to understanding root water dynamics or the electrical methods being used.

Sentence rephrased to explain the concept of current leakage defined by Peruzzo et al. (2020): *"Peruzzo et al. (2020) hypothesize that drought stress can also reduce electrical current leakage wherein the current exiting the plant root at the proximal part is decreased, particularly for woody species."*

Ln 132: this is the first use of EC abbreviation which has not previously been described but is used through page 6

All occurrences to EC were substituted by impedance to take into consideration reviewer 2 comment i.e. "The signal that is measured is also related to the electrical resistivity or conductance of the root tissue and of the soil and not only by the capacitance." Impedance combines both capacitance and resistance.

Ln 207: Rephase, i.e. For each irrigation event we regulated the amount of water supplied based on the information obtained from the scale data. The plant received 75% of the measured transpiration since the last irrigation cycle.

Sentence rephrased "*We controlled the water supply for each irrigation event based on the data obtained from the scale, ensuring that the plant received 75% of the measured transpiration accumulated since the last irrigation cycle.*"

Ln 214: in Table 1 the cycles go from May 13th to July 12th

Corrected using Table 1 datetimes.

Ln 219: Cycle 9 in Table 1 says it uses holes H1-H8 and is not colored green or orange which conflicts with the statement here that 'From cycle number 3 to 9, we restricted the water input to the two lateral holes'. Also, although lateral does mean coming from the sides, it is not often used i in that context so I would consider changing it to something more descriptive, like left-most and right-most.

"Lateral" changed into "left-most and right-most"

Table 1: see Major comment 1 above

See response above

Ln 233: an abbreviation for electrical resistivity is defined on Ln 50 but then only used intermittently - make this more consistent throughout

Electrical resistivity was replaced by ER for all occurrences except the first one.

Ln 301: the abbreviation given here is ICSD, although subsequently only CSD is used

Corrected. ICSD abbreviation is now replaced by CSD inversion.

Ln 341: 'were' is typed twice in this sentence

Corrected thanks.

Fig 2b: the time series has a datetime mix up. Labelled August 6 (8/6) instead of June 8 (6/8). It would be helpful to remind the reader in this paragraph (~Ln 351) that the measurements shown come from the 26 leaves which is described back in the methods.

Figure label corrected and sentence rephrased: "*The measurements shown come from the 26 leaves (c.f section 2.5) and indicate that the plant is under high water stress at the end of the irrigation cycle (one week after the last partial irrigation, on June 8,2022), and under lower water stress one day after irrigation (on June 16, 2022).*"

[Figure]

Fig 3: this transpiration data is really nice to see

Glad that you like it and hope to see it from other related studies again.

Ln 380: I would change this to say 'Fig 4 shows the trend for the irrigation cycles (-1 - 8) since cycle -1 was not PRD and cycles 0 and 1 were not the same as cycles 3-8. Also, here and in Fig 4, Cycle 9 is dropped, which conflicts with Ln 219 and Table 1.

Changed were made accordingly.

Ln 405: Reword to specify which side is the 'irrigated side'. Also, when you say the ER of the irrigated side had dropped by 20% how is that being calculated?

Sentence rephrased: "*One hour afterwards (+ 1H) the ER of the left irrigated side had dropped by 20% (estimated from the averaged values spanning from the middle of the rhizotron to the left boundary).*"

Fig 5: see Major comment 1

Changed were made accordingly.

Ln 426-427: It's not immediately clear why this confirms the quality of the estimated background values. Can you elaborate on what would be expected here and the underlying mechanism?

For the soil injection, considered punctual, we assumed that one single current source is responsible for the entire voltage distribution. Nevertheless, as the current is modulated by the

soil ER, a bias on the latest would create an error in the forward current source imaging and ultimately in the position of the current source. The soil CSD result showed that is always pinpointed to the location of the injection electrode whatever the irrigation pattern which is why we consider that our ER model is of good quality. More details can be found in (Mary et al., 2020)

We added a sentence to explain this:
"*Considering the modulation of current by soil electrical resistivity (ER), any bias in ER could introduce errors in forward current source imaging and, consequently, affect the positioning of the current source.*"

Fig 6: Rescaling the y axis on the center of mass to be narrower would help accentuate any slight variations. This graph should also have a unit (cm?). The caption mentions cycles 5 and 6 were used in Fig 7 but based on the dates it is actually cycles 6-7 (see my point in Major comment 1.3.3 above).

Ok figure improved (scale + unit) and legend corrected

[Figure]

Ln 457: 'in' is typed twice

Corrected thanks.

Lns 460-47: it would be helpful to explicitly state how mean SWC was calculated for each side (i.e. all nodes left of center averaged). Were nodes on the edges excluded?

We added a sentence to explicitly state how mean SWC was calculated: "*Averaging is performed on the mesh nodes falling within each side, with the middle point being defined as half of the rhizotron width, equivalent to 0.25m.*"

Ln 499: The dates in Fig 3 and Table 1 suggest that the decrease in the rate of uptake is happening between July 5th and July 11th between cycles ⅞

We are not sure why the reviewer considers that the rate of uptake is happening between July 5th to July 11th. Figure 3 shows that the curve of the measured weight flattens after July 11th i.e. for the last cycle as stated correctly in the manuscript.

Ln 537:539: consider rewording to use less colloquial language.

Sentence rephrase: "*In our experiment, the first criterion was met, but not the second. This provides an interesting piece of evidence, leading to the following considerations.*"

Ln 540:549: I like this discussion point regarding the potential impact of root growth over the course of the experiment

Thanks.

Ln 573: I would rephrase 'really good' to something more specific or to simply 'good'

Ok done

Ln 575-577: I'm finding this sentence unclear, please rephrase

Sentence rephrased: "*The algorithm has undergone testing in a rhizotron experiment and has demonstrated the ability to differentiate punctual sources, even when their current contribution is as low as 5% of the total current (Peruzzo et al., 2020).*"

Ln 581: 'the' is typed twice in this sentence

Corrected, thanks.

---

## Author Comment (AC2)

In this paper, electrical resistivity tomography and electrical current imaging are used to monitor water content distributions and the distribution of electrical current from the root system into the soil. The experiment is carried in a rhizobox, so that transpiration rates and total soil water contents could be monitored carefully. The water application is alternated between the two different sides of the box and is changed over time to generate a certain stress level in the second part of the experiment.  To my opinion, the most interesting outcome of the study is that the electrical source when an electrical current is injected in the plant shoot, is apparently more homogeneously distributed (one could assume homogeneously distributed along the root length) when stress occurs. Another important result is that with ERT, the water content changes over time could be imaged quite accurately.

In the following our response to the reviewer's comment is written in blue.

We express our gratitude to the reviewer for providing valuable and constructive feedback on our work and for highlighting the interesting outcomes. We have carefully considered and addressed all of the reviewer's comments, which we believe have significantly enhanced the quality of the manuscript. We were particularly impressed by the reviewer's suggestions in interpreting the results and we would be happy to cite his review directly in the manuscript.

The main changes we have made include:

(i) improvement of figures quality (consistent colours), proofreading the text for typos and rephrasing it when necessary.

(ii) Modifying the description and discussion of our experiment by replacing "PRD" with "root water limited availability," as indicated in the revised title. This change had only a minor impact on the actual results, as they were already aligned with.

(iii) Responding to the reviewer's comment by incorporating new references in the introduction that briefly cover various methods used to measure root physiology, anatomy, and biomass. By discussing the assumptions associated with these methods, we are able to establish up-to-date parallels between current pathways and water pathways.

(iv) Completely reshaping figures 8 and 9 and discussing the observed current density in this experiment, considering it as a result of both higher transpirational demand and/or drier soil conditions.

These revisions and improvements have enhanced the manuscript, and we sincerely appreciate the reviewer's insightful comments that have contributed to its overall quality.

A first main comment on the paper is that the authors describe their experiment as a partial root zone drying experiment. But, in such type of experiments, a part of the root zone is continuously kept wet by nearly continuous application of water to a part of the root zone using for instance drippers, whereas the other part is left to dry out. In their experiments, water is also applied to a

part of the root zone and the application is alternated between the two sides. But the duration between the applications, is quite long so that both parts of the root zone dry out to the same level and water can flow from one part to the other. These conditions do not really generate a spatially variable root water uptake. The authors correctly recognize that their experimental setup did not exactly reproduce partial root zone drying experiments. To avoid confusion, I would propose not to call the experiments PRD experiments.

The reviewer's assessment is accurate: our intended "PRD" technique does not selectively stress half of the plant but rather imposes stress on the entire plant. This aligns perfectly with the results, which indicate that the plant attempts to extract water from both sides due to its lack of knowledge about the water source or its origin. During our discussion, we acknowledged that our experimental setup did not precisely replicate partial root zone drying experiments, as pointed out by the reviewer. Consequently, we addressed this main comment by adjusting the terminology; specifically, we replaced all instances of "PRD" with "limited water availability."

The title was rephrased (also considering the comment by Reviewer 1).
"*Imaging of the electrical activity in the root zone under limited water availability stress: A laboratory study for Vitis vinifera.*"

A second main comment is on the relation between soil water content changes and root water uptake and the interpretation of current source distribution images. Due to water redistribution in soil, local soil water content changes must not be interpreted as local root water uptake. In the text, the authors seem to allude to this although they do not use this approach when interpreting the electrical resistivity images.

We acknowledge the reviewer's comment and have revised our interpretation of root water uptake (RWU) by taking two actions:

- We added a paragraph into the introduction to develop this concept:
  "*The correlation between root water uptake and soil water content changes exists when averaged over a larger spatial scale than the scale at which soil moisture redistribution can compensate for local root activity. The determination of these spatial scales depends on the soil's hydraulic properties. The correlation between root water uptake and changes in soil water content can also be influenced by the time scales in addition to spatial scales. The ability to discriminate between them relies on factors such as the soil hydraulic properties, rates of local water extraction, and the temporal dynamics of water redistribution in the soil (Anonymous Reviewer, 2023)*\**"
- The results and discussion of the manuscript were carefully reviewed to ensure that local fluctuations in soil water content were not mistakenly attributed solely to local root water uptake.

\*Anonymous Reviewer: Comment on bg-2023-58, https://doi.org/10.5194/bg-2023-58-RC2, n.d.

Concerning the current source distributions, there it would be helpful if the authors could make the analogy between water and electrical current flow in the soil root system. The distribution of both depends on the distribution of the water and electrical conductivities in the soil and the plant/root system. In unsaturated soil, both conductivities change with the water content and they also change very close to soil-root interface so that the conductances close to the soil-root interfaces may differ considerably from the bulk soil conductances, The latter changes depend strongly on the local flow densities near the root surfaces and hence on the transpirational demand. To my understanding, the results that are presented (how the current density distribution changes when water stress occurs, which is mainly the result of a higher transpirational demand and not of drier soil conditions in this experiment) suggest that these changes in conductances of both water and electrical conductances near soil-root interfaces may explain these observations.

We agree with the reviewer.

Gradients in matric potential occur between the bulk soil and the soil–root interface once plants start to transpire. During soil drying, how quickly and how far $\psi_{soil-root}$ deviates from $\psi_{soil}$ with increasing transpiration depends on soil textures and root hydraulic phenotypes.

To give a hint, we made a tentative conceptual figure to understand the analogy between water and current flow in the soil root system based on and possible use for our experiment specificity i.e. knowing that both soil water and transpiration rate could have affected the current pathway. The figure is inspired from the following papers:
- (Cai et al., 2022), figure 1
- (Doussan et al., 1999) figure 1
- (Manoli et al., 2014) figure 1
- (Couvreur et al., 2012)

We added a sentence in section 1.1 to explain the conceptual figure "*According to Fig.1, the gradient Δψsoil = (ψsoil- ψsoil-root) is higher in dry soil than in wet soil. The soil conductance gs is equal to the evapotranspiration E divided by Δψsoil, and thus gs increases when the soil dries and E remains constant. The same occurs for the root conductance gr . The root axial water flow rates Qx (L3T−1) and root radial water flow rates Qr (L3T−1) can be solved analytically by solving the system of equations of Ohm's and Kirchhoff's laws (Couvreur et al., 2012).*"

[Figure]

Figure 1: Conceptual figure showing the position of the plant in the rhizotron. The water input was done alternatively from left (a) to right (b) via small holes on the top of the rhizotron (H1 to H8). The roots are free to grow on both sides of the rhizotron. The circles on the screening face show the locations of the electrodes. Two additional electrodes (needles) are used for the ECI, one for the stem injection and the other for the control soil injection next to the stem. The rhizotron is weighted by a central point load scale (PC60-30KG-C3, Flintec) mounted between two support plates in plexiglass. The line below describes the state of the art of hydraulic conductivity at a single root and the distinction between dry (c) and wet (d) soil. The figure draws inspiration from the electrical circuit analogy of RWU (Root Water Uptake) proposed in previous works (Doussan et al., 1999, Manoli et al., 2014 and Couvreur et al., 2012). In a recent article, Cai et al. (2022) schematized the gradient of potential ψsoil, ψsoil-root and ψroot, along with the corresponding hydraulic conductances of the soil, the soil-root interface, and the root (represented as gs, gsr, and gr, respectively), in response to high or low transpiration demand (E). Note that the soil-root interface and the xylem cell interfaces are seats of current polarization due to the formation of the Electrical Double Layer (EDL) well described in Tsukanov and Schwartz (2021).

Along the same lines of reasoning, I do not think that under conditions when soil electrical conductances are high near the soil root interface and when there is good electrical contact between soil and roots, current source density distributions are related to water uptake distributions.

We agreed with the reviewer's opinion. Nevertheless, given the current state of the art, there is no demonstration of this given the fact that up to now the relationship between water and electrical conductances has not been firmly established. This is clearly something that we wanted to address in this article, particularly from the original Fig. 9.

A third main comment is that the interpretation presentation of the results is often not clear to me. For instance, figure 8 seems to suggest that resistivity increases with increasing water content. This is opposite to what is generally known and opposite to Archie's Law. At several points, I could not follow the reasoning of the authors and the text should be proofread carefully. I noted a few spelling errors but these are not exhaustive.

We acknowledged that Figure 8 contained evident errors caused by inaccurately interpolating weight data during the ERT acquisition time. In response, we made the decision to replace Figure 8 with a simplified comparison, demonstrating a direct 1:1 correlation between the variations in soil water content inferred from the scale and those derived from the ERT. Further information and detailed explanations can be found in our response to question Ln454.

Below are detailed comments that were written during a first read of the paper. They reflect my confusion that sometimes occurred when reading the paper.

Introduction: The introduction part on the electrical capacitance and electrical current imaging should be clearer by better stating which assumptions are made in these methods and which root traits and soil properties could (potentially) be derived from these methods. For instance, in capacitance imaging, a lumped property of the root system is derived. But it was not immediately clear to me how that is done and what the underlying assumptions are. Which assumptions about the axial and radial electrical conductances of the root system are made and how do these properties determine the total root system capacitance? It would be helpful if an

analogy to root system properties that relate to water flow could be made, like root system conductance (root system capacitance for water flow is generally not considered).

We respectfully disagree with the reviewer's comment as the assumptions regarding the radial and axial electrical conductances were clearly stated, and their influence on the estimation of the total system capacitance was extensively documented with multiple references.

Another issue is that is not clear to which electrical properties of the root system is referred to. I think both capacitance and resistance should be considered. Finally, the abbreviations used are confusing: ECroot stands for capacitance of the root system, ECI for electrical current imaging. Note that EC is often used for electrical conductivity.

We added a new paragraph in the introduction (1.1) to recall the main application range and variations of stem-based methods:

"*There is a variety of methods used in the literature with applications ranging from biomass estimation, root morphology to root physiology (root activity). At a single frequency, we distinguish between ECM methods which rely on capacitance measurements and are commonly used to study root systems at the plant scale, and EIM, which measures both capacitance and resistance. Capacitance represents the polarization processes and measures the charges stored during the current flow. Both use the fact that the root can polarize at the soil-root interface and inside the root to infer direct root-related information such as dry and wet mass, surface area,...). A second group of methods Electrode Impedance Spectroscopy (EIS) uses a range of frequencies to capture the polarisation processes sensitive to the root physiology and anatomy. For a detailed description of the methods, the reader is invited to refer to (Ehosioke et al., 2020).*"

ECroot is now written fully as "root electrical capacitance" to avoid any confusion with ECI.

Ln 32 'The partial root zone drying (PRD) method is part of an ensemble of irrigation strategies that aim at improving water use efficiency. It consists of irrigating only one part of the root system of the same plant using a certain percentage of the potential evapotranspiration (ETp), usually inferior to the total water needed.' I would be great if you could include some explanation about the difference between partial root zone drying versus deficit irrigation and why a partial drying of the root zone would lead to a better result than drying of the entire root zone. In fact, as it turns out later, in the experiment that was conducted, the whole root zone dried out during a drying cycle.

We added a sentence in the introduction to mention the differences in the pros and cons of PRD VS DI.

"*Under conditions of high evaporative demand, both PRD (Partial Rootzone Drying) and DI (Deficit Irrigation) led to increased stomatal closure. This reduction in stomatal conductance in*

*response to soil water deficit is attributed to the production of abscisic acid (ABA) in roots, triggered by the drying soil (as reviewed in Loveys et al., 2000; Davies et al., 2002). Notably, if there is adequate sap flow through the roots, the ABA signal is transmitted through the xylem to the leaf, as demonstrated by Dodd et al. (2008). According to Davies and Hartung (2004), it is proposed that plants subjected to partial root-zone drying (PRD) demonstrate improved performance compared to plants under deficit irrigation (DI) when an equal amount of water is applied. This is attributed to the ability of PRD to stimulate root growth and maintain consistent signalling of abscisic acid (ABA) to regulate shoot physiology. Davies and Hartung (2004) stated that the effects of PRD on plant growth, yielding and functioning are quantitatively different from those of RDI. One of the advantages of PRD when operated properly, is that plants sustained and even increased shoot and fruit turgor even though a reduced amount of water is applied to roots (Mingo et al., 2003). On the other hand, one of the disadvantages of RDI is that the entire root zone is allowed to dry out, the roots can become stressed and damaged and if not rewetted can die and signalling may diminish. Conversely Fernández et al. (2006) stated that not always a PRD treatment has been found advantageous as compared to a companion regulated deficit irrigation (RDI) treatment and demonstrated it in a study on olive trees in which sap flow measurements, which reflected water use throughout the irrigation period, showed no evidence of stomatal conductance being more reduced in PRD than in RDI trees. Collins et al. (2009), in an experiment on the grapevine (Vitis vinifera L.) show that the response to PRD applied at 100% ETc and deficit irrigation applied at 65% ETc was the same, increasing stomatal sensitivity to vapour pressure deficit and decreasing sap flow."*

Ln 35 'Application of PRD triggers a physiological response in the plant via a hormone called Abscisic acid (ABA), which is produced in the roots and transmitted to the leaves to regulate the stomata closure and thus reducing water transpiration while keeping photosynthesis active and finally leading to increased water use efficiency.' What is different compared to entire root zone drying? Why is it important to dry out only a part of the root zone?

See the response to the previous question. We also added a sentence to stress the importance of soil study in understanding reduction in stomatal conductance:
*"According to (Cai et al., 2022), while stomatal conductance is a significant above ground hydraulic factor influencing water use in crops, it should not discount the role of belowground hydraulics, as changes in soil-plant hydraulic conductance have been found to drive stomatal closure (Abdalla, Carminati, et al., 2021). This highlights the crucial importance of studying electrical activity in the soil."*

Ln 48: 'soil moisture patterns determined by PRD are visible from the ERT perspective and can be attributed to the root system distribution.' What do you mean by this sentence? What is meant specially by 'attributed to the root system distribution'? Do you mean that the pattern of drying in the part of the root zone that does not receive water can be related to the distribution of the roots in this zone?

Thanks for spotting this. Sentence rephrased:

*"The observed drying pattern resulting from an elevated evapotranspiration rate (ER) in the non-irrigated section of the root zone matches the root distribution in that area, while the observed wetting pattern arising from a decreased ER in the irrigated section of the root zone can be attributed to the irrigation itself."*

Ln 50 'Roots induce changes in the soil structure in terms of porosity and hydraulic conductivity which ultimately modify the water pathways and fluxes and thus the ER itself.' This is certainly correct but isn't the question to what extent this is a secondary effect compared to the primary effect that is caused by the uptake of water by the roots?

The reviewer's observation is valid: alterations in soil structure may have a lesser impact on Electrical Resistivity (ER) compared to root water uptake (RWU). However, it should be acknowledged that this relationship may not hold true for species with extensive root systems, such as woody species. This aspect also intersects with the discussion on water redistribution and channelling. During rainfall or irrigation events, the ER in the root zone can undergo significant changes, influenced by the root anatomy which varies among different root systems. Further investigation is necessary to gain a deeper understanding of the intricate interplay among soil structure, RWU, and ER in this context.

One sentence was added:
*"Soil structure changes may have a relatively smaller effect on ER than root water uptake RWU, although this may differ for species with extensive root systems like woody species; this is further true during rainfall or irrigation considering water redistribution and channelling influenced by varying root anatomies and causing dynamic variations in ER."*

Ln 90: 'appoplastic' should be 'apoplastic'

Corrected thanks

Ln 97 'complex balance between reducing radial flow (as a consequence of ABA signaling sent by the roots) to conserve water in the soil but keeping the axial flow active.' I am not following the reasoning here. How can axial flow be kept when radial flow is blocked? The reason for reducing the radial flow in dry soil is not to conserve water but to avoid too strong water potential drops between the soil and the plant. By reducing radial conductivity in dry regions, plants shift the uptake towards wetter regions where the soil conductivity is higher so that plants can take up water at the same rate but keeping higher plant water potentials.

The reviewer is right this can be confusing and we rephrased the sentence and added a reference.
We meant that there is a trade-off between radial and axial flow. Obviously, at a single root scale, the axial flow cannot be kept when the radial flow is blocked but there could be a balance. We replaced balance by tradeoff. Aroca R (2012)* describes in a generic manner the plant responses to drought stress. Furthermore, we think that the tradeoff is what the suberization

induces i.e. it keeps a good longitudinal conductance but limits the radial one.
*Aroca, R., Porcel, R., and Ruiz-Lozano, J. M.: Regulation of root water uptake under abiotic stress conditions, Journal of Experimental Botany, 63, 43–57, https://doi.org/10.1093/jxb/err266, 2012.

Ln 114: 'Without being able yet to give hints about the electrical current pathway, recent advancements in the development of explicit RWU models, based on plant hydraulics, provide insights into how robust capacitance models hold and under which conditions. We learnt, for instance, that at the root level, RWU models account for the anisotropy by separating the root hydraulic conductance into two terms (longitudinal and radial).' I think the authors should give references here to explicit RWU models and also refer to work that used these models to simulate electrical currents (and polarization) in root systems (see for instance work by Mathieu Javaux and colleagues and Nimrod Schwartz and colleagues).

We cited (Javaux et al., 2008; Couvreur et al., 2012). As for the work from Weigand, 2017; Weigand and Kemna, 2019; Tsukanov and Schwartz, 2020, 2021 they are more centred around polarisation than RWU so It was mentioned directly in the introduction.

Ln 119: 'Up to now the relationship between root water content and root hydraulic conductivity with electrical resistivity has not been firmly established. Many other parameters can affect the water flow as well as the current pathway of stem-based methods.' This is quite vague, especially in view of work done by others previously. Which 'other parameters' are you referring to?

Many "other parameters" are now replaced by "root function, age,  water retention capacity and transpiration rate in particular (Ehosioke et al., 2020)"

Ln 132: I suppose you are referring here with EC to electrical capacitance. This may be confusing for many readers since EC is typically associated with electrical conductivity. I am also wondering why you use capacitance and not impedance, which combines both capacitance and resistance. I suppose the signal that is measured is also related to the electrical resistivity or conductance of the root tissue and of the soil and not only by the capacitance.

Initially, we used EC to relate to electrical capacitance as the studies cited are using this terminology. Nevertheless, we agree with the reviewer's suggestion that impedance would be more appropriate here. All occurrences of EC were substituted by impedance.

Ln 151: 'we aim at showing that the current path through the root system is linked to the active root zones.' Doesn't this imply that it is assumed that soil and root hydraulic conductances are

positively correlated to electrical conductances?

The reviewer's observation is accurate, from a current pathway or hydraulic conductance perspective, the assumption of a positive correlation between soil and root hydraulic conductances and electrical conductances yields similar outcomes. However, it is important to note that in the introduction (ln114 to 121), we explicitly stated that the relationship between root water content, root hydraulic conductivity, and electrical resistivity lacks definitive evidence. Consequently, establishing this relationship constitutes one of the key objectives of our study.

To enhance clarity, we have reformulated our aims to make them more explicit, particularly for readers who prefer to approach the topic from a hydraulic perspective.

BEFORE:
"*The aim of this study is twofold:*
*(i) we aim at showing that the current path through the root system is linked to the active root zones*
*(ii) …*
"
AFTER:
"*The aim of this study is twofold:*
*(i) we aim at showing the correlation between the current path through the root system and the active root zones. This assumption is based on the notion that soil and root hydraulic conductances are positively associated with electrical conductances.*
*(ii) …*
"

Ln 158 'changes in soil water content measured by ERT are a relevant spatial proxy of root activity' It has been discussed in several papers that changes in soil water content do not map to distributions of root water uptake or root activity. Local root water uptake can be compensated by water redistribution in the soil and decouple local water content changes from root water uptake. Maybe it is better to write that root water uptake and soil water content changes that are averaged over a spatial scale that is larger than the scale over which water redistribution in the soil can compensate soil moisture changes due to local root activity, can be correlated. These spatial scales depend on the soil hydraulic properties and the local extraction rates.

(Repeated response from main comment #2)

We acknowledge the reviewer's comment and have revised our interpretation of root water uptake (RWU) by taking two actions:

- We added a paragraph into the introduction to develop this concept:
  "*The correlation between root water uptake and soil water content changes exists when averaged over a larger spatial scale than the scale at which soil moisture redistribution*

*can compensate for local root activity. The determination of these spatial scales depends on the soil's hydraulic properties. The correlation between root water uptake and changes in soil water content can also be influenced by the time scales in addition to spatial scales. The ability to discriminate between them relies on factors such as the soil hydraulic properties, rates of local water extraction, and the temporal dynamics of water redistribution in the soil (Anonymous Reviewer, 2023)\*"*

- The results and discussion of the manuscript were carefully reviewed to ensure that local fluctuations in soil water content were not mistakenly attributed solely to local root water uptake.

Ln 162: 'during the application of PRD, only one part of the root system would be active and the current injected in the stem would preferably spread to the side where the root system is irrigated.' This assumption seems to be contradictory in itself. Partial root zone drying only occurs when the part of the root system in the region that does not receive extra water can remain active for a while. So I think it is better to write, ' When during the application of PRD, the part of the root system in the dry zone is deactivated, current injected in the stem would preferably spread to the side where the root system is irrigated.'

Ok, sentence rephrased: "*During the implementation of root-zone limited water availability when a portion of the root system in the dry zone becomes deactivated, injected current in the stem tends to preferentially propagate towards the side where the root system is irrigated.*"

This assumption hinges on the assumption that a deactivation of the root system part in the dried out zone corresponds with an increase in root and/or soil electrical resistivities. That electrical resistivities of soil increase with soil drying is trivial. But, the question is whether electrical resistivities of the coupled soil-root system increase to the same extent with soil drying as the hydraulic resistances and decreasing soil-plant hydraulic potential differences. I suppose this hypothesis will hold true in coarser soils but for clay soils, this can be questioned.

The reviewer is right pointing at the lack of definitive evidence for the hypothesis. Yet one may consider the analog of classical contact resistance between e.g. metal electrodes and soil, that also increases as soil resistivity increases (e.g. when the soil gets drier).

Figure 1: give indications of the height, width and depth of the rhizobox in figure 1.

The experiment was conducted using a rhizotron 50 cm wide, 50 cm high, and 3 cm thick. We added it to the figure directly as suggested.

Ln 180: 'An outlet point was placed on the bottom right side (z=5cm) and the rhizotron was always saturated below this point. In the course of the experiment (after the growing period) no water discharge was observed through the outlet point.' How was the bottom of the rhizobox

kept saturated? Was the outlet connected to a Mariotte system? Was regularly water added? If no water was added at the bottom, then I wonder why the bottom remained saturated.

No, a Mariotte system was not employed in our study. The visible observation from the screening face of the rhizotron, where the soil below the outlet point appeared saturated, can be attributed solely to the outlet point being positioned above ground level. The only possible ways for the soil to desaturate would be through soil suction or root water uptake (RWU), but visually, this was not observed.

Ln 295: '(2) time-lapse inversion (difference inversion) where the difference in resistivity is inverted between a given survey and a background survey (in this case, the background survey is the previous one).' Here it is important to give the time difference between the two measurements. In order to be interpretable, the time difference should always be the same. Were daily measurements taken always at the same time of the day?

Daily measurements were not consistently conducted at a fixed time, varying between 12:00 and 18:00 when the light is on. However, the time interval between irrigation and subsequent measurements remained consistent for each cycle with H+1 and D+6. To provide clarity, we included the irrigation time in Table 1 and the figure header.

Ln 417: 'Time steps correspond to measurements before (a), after one hour (b) and after 6 days (c).' I do not understand well how the differences are calculated and what the time step of the differences are. Was every day measured at the same time or at different times? Was the difference calculated between the measurement at a certain day and the day before it or was it calculated from the difference between the measurement and the measurement at the start of the irrigation cycle.

Based on the feedback from reviewer 1, we acknowledged that the figure legend was causing confusion. In response to this, we made the necessary changes to the table header to provide clearer indications of the time steps. Additionally, we extended these modifications to the figures included in the Supplementary Material to ensure consistency and facilitate better comprehension.

| (a) Background (-4h) = 2022-06-29 9:30 | (b) Just After Irrig. (+0h15) = 2022-06-29 14:15 | (c) 6 days after Irrig = 2022-07-05 16:35 |
|---|---|---|

[Figure]

Figure 5: Spatial distribution of the resistivity (in Ωm) and changes (in %) in ER obtained by a time-lapse inversion for cycle 8 following partial left irrigation of the rhizotron (2022-06-29 13:45-14:00, 386 ml). Time steps correspond to measurements before (a), -4h min (b) just after irrigation (+0h15) (c) and 6 days after irrigation started.

Ln 454: 'Figure 8 shows the relationship between the variation between two consecutive measurements of the weights with the variations of average electrical resistivity (Fig.8a, R2=0.76, p-value=6.5 x 10-5) and those of resistivity-derived average water content (from Archie's law - Fig.8b, R2=0.815, p-value=6.8 x 10-6). An increase in weight over time is positively correlated with an increase in resistivity and water content meaning that the changes in resistivity are mainly associated with transpiration (rather than changes in soil structure or other parameters).' I do not understand these results.

You are plotting changes in weight over time versus resistivities or versus water contents. Shouldn't changes in weight be plotted versus changes in water content or changes in resistivity? Although I am not sure whether the latter makes sense since water content and resistivity are non-linearly related in Archie's Equation. But, what amazes me the most is that resistivity increases when the weight and hence the water content increases. This cannot be correct. Finally, from the changes in weight, a change in water content can be calculated. These changes in water content calculated from weight changes should be one-to-one related to the changes in water content calculated from the ERT measurements. I propose comparing those.

In a new figure, we plotted the change in water content inferred from the changes in weight versus the change in water content inferred from the ERT after Archie's transformation (see figure below). As expected this is a 1:1 relationship with consistent negative variations of SWC for both ERT and scale during irrigation time and positive variations during RWU/transpiration times.

[Figure]

In the revised manuscript, new results are described:

*"By examining the fluctuations in weight, one can calculate the corresponding changes in spatially averaged water content. Figure 8a illustrates a linear trend (R2=0.83 and p=2.96e-6) between the inferred water content variations from the scale and those obtained from ERT (after Archie transformation). The most significant negative changes in averaged water content are attributable to the triggered irrigation, leading to a ΔΘ (change in water content) of -0.1. Conversely, positive changes primarily result from transpiration, with a maximum value located at +0.1."*

Ln 484: 'For cycles where stressed was not applied (i.e. < cycle 3), for the stem injection, Ns1 is distributed between 5 and 25%.' What is remarkable is that the distribution of sources is not related to the soil water content. At low water contents, when one would expect more stress, the distribution can be like the distribution for the soil injection or like the distribution when water stress is supposed. But also at high water contents, the distribution can be like the distribution when water stress is supposed. Can the grey triangles that are falling in the range of the black triangles be explained? Could they correspond with higher transpiration rates. It seems to me that the stress is related to the transpiration rate or transpiration demand which increases over time due to an increase in leaf area. At high transpiration demands or rates, stress may occur at higher soil water contents because then the soil becomes limiting for the root water uptake.

We replotted the figure by considering the time variation of Ns (Number of sources which carry at least 1% of the total current) (see Fig. 8b below) and demonstrated indeed that the distribution was not linked to the soil water level as there are no significant differences between before and after irrigation, but that Ns increase with the transpiration demand (via an increase of the leaf area index LAI).

[Figure]

In the revised manuscript, new results are described:
"*Figure 8b shows the relationship between the variation of the percentage of the current sources carrying at least 1% of the total density (Ns1) used as an estimator for current density dispersion with respect to the datetime of the experiment. For the soil injection (red dots), Ns1 is relatively constant between 5 to 10% of the total number of possible injection nodes (grey area). For the stem injections, Ns1 increases over the course of the experiment. From June 1st to July 8th, the Ns1 triple. The is no distinction between Ns1 measured before (triangle point) and after (crossed points) irrigation*".

and discussed:
"*Based Fig. 2 and 8b, the association between water stress and leaf development, along with transpiration demand, is expected to be more prominent (and increasing during the course of*

*the experiment rather than the specific time points before and after irrigation). Indeed the fluctuations in water content during various cycles, with or without stress, exhibited remarkable similarity. Both stressed and non-stressed cycles experienced a drop in water content to similar low levels. Consequently, water content does not appear to account for the variability in water stress. Instead, it is the increased transpiration demand over time that seems to play a more significant role in driving the observed changes. At high transpiration demand, stress may occur at higher soil water contents because the soil becomes limiting for the root water uptake. The changes in water potential and water content in the vicinity of the soil-root interface can potentially impact the electrical conductivity of the immediate soil surrounding the roots. Consequently, as the experiment progressed, lower electrical conductances in the soil around the roots, potentially led to a restriction in the flow of current between the root system and the soil. This, in turn, may have resulted in a more uniform distribution of the electrical current source along the entire length of the root system."*

Ln 501: Garre et al. 2011 is not in the reference list.

Well spotted, thanks we added it to the reference list.

Garré, S., Javaux, M., Vanderborght, J., Pagès, L., and Vereecken, H.: Three-dimensional electrical resistivity tomography to monitor root zone water dynamics, Vadose Zone Journal, 10, 412–424, https://doi.org/10.2136/vzj2010.0079, 2011.

Ln 505: 'Our observation is in line with the literature i.e. in general, low soil water content (SWC) can lead to drought stress in plants, which can result in decreased leaf stomatal conductance and less transpiration, and vice versa.' Actually, I think that your observations in fact show the opposite. The water content changes during the different cycles with or without stress were very similar and water contents dropped to the same low levels in cycles where no stress was observed as in cycles where stress was observed. Therefore, water content does not seem to be explaining variable for water stress. It rather seems to be the transpiration demand which increased over time.

We share the reviewer's perspective on the matter.

While it is true that stress levels escalated due to the increasing transpiration demand over time, it is worth noting that there were discernible differences in water stress at the conclusion of the irrigation cycle and one day after irrigation (on June 16, 2022) based on leaf stomatal conductance measurements, albeit with smaller variations.

In light of this, we incorporated a sentence to highlight that, overall, water stress appeared to be more closely linked to leaf development and transpiration demand rather than the specific before/post irrigation time points.

*"Based on Fig. 2, the association between water stress and leaf development, along with transpiration demand, is expected to be more prominent (and increasing during the course of the experiment) than the specific time points before and after irrigation."*

Ln 518: 'This is a hint that the hydraulically stressed plant tends to have a wider and deeper active root system, even not necessarily active only on the side where the PRD is temporarily applied. Possibly the reaction of the plant to the changing side is too slow to show up in our measurements, but the reaction to general stress is apparent.' I am sorry but you lost me here.

In this section, we focus on the initial stages of the experiment when stress was minimal, and the hydraulic conductance of the soil-root interface was likely high. The discussion concerning the current density distribution in relation to stress is presented later in section 4.2.

The sentence was rephrased:
*"Our observations potentially suggest that under conditions where soil electrical conductances are high near the soil-root interface and even if there is good electrical contact between soil and roots, the distribution of current source density might not be directly related to water uptake distributions. Further research is needed to confirm this potential relationship."*

Ln 532: 'tend to show that mixed soil-root pedophysical relationships are preferable (e.g. Rao et al., 2018).' I think that rather than considering mixed soil-root pedophysical relations, it would be important to consider small scale variations around single root segments in water content and /or soil hydraulic properties.

The reviewer is right. Both concepts are related as mixed soil-root pedophysical relationships try to introduce knowledge about the soil-root relationship. We added a sentence to explicitly consider the reviewer's comment:
*"Moreover, considering small-scale variations around individual root segments in terms of water content and soil hydraulic properties becomes crucial for a comprehensive understanding of the system."*

Ln 552: 'Additionally, capillary rise may have taken place due to the presence of a saturated zone at the bottom of the rhizotron' Yes, but then the water content at the bottom of the rhizobox should decline over time since no water was added at the bottom of the box?

See previous answer Ln 180. This statement has nevertheless been moderated by rephrasing:
*"Additionally, even not visible from the screening face capillary rise may have taken place due to the presence of a saturated zone at the bottom of the rhizotron"*

Ln 559: 'Given the stress applied, the ER changes highlighted that root played a major role in the wine plant survival and evidenced strategies of adaptation. Indeed, the plant was able to change its water uptake zones depending on the water availability, from all places, not only from

the alternate irrigated areas.' I don't think this conclusion can be drawn from this experiment. Plant adaptation means an active adaptation of the plant to redistribute the water uptake.

- First, I am not sure whether uptake distributions were directly observed.
- Second, also without any adaption, the uptake distribution changes when the soil water content distribution changes, but also when the water uptake rate changes. The impact of water potential distribution on uptake distribution is trivial. The water uptake rate or transpiration rate may impact the uptake distribution since the soil water potentials near the soil-root interface will drop which leads to a drop in soil conductivity. This dependency on water potentials and flow rates make that the conductivity distributions in the soil-plant continuum, and hence the uptake distributions change with flow rate. I suppose that the change in water potential and water content close to the soil-root interface also had an effect on the electrical conductivity of the soil just around the roots. This means that later during the experiment, the electrical conductances in soil around roots were lower and might have become limiting the current between the root system and the soil. This would have generated a more homogeneous distribution of the electrical current source along the total length of the root system.

Regarding the first reviewer's concern, we only added and/or water redistribution to make sure that RWU is not the only strategy of adaptation for the plant to stress.

Sentence rephrased: "*Given the stress applied, the ER changes highlighted that root played a major role in the wine plant survival and evidenced strategies of adaptation. Indeed, the plant was able to adjust with water uptake **and redistribution** zones depending on the water availability, from all places, not only from the alternate irrigated areas.*"

We already responded to the reviewer's "second" issue in the previous question (Ln 484) regarding the effect of the changing electrical conductances in the soil around roots and its consequences in terms of current distribution.  We express our gratitude to the reviewer for providing us with their scientific perspective, which helped us explain the changes in current distribution over time. We have incorporated their valuable insights into our revised discussion.

Ln 612: 'We only evidenced that the Current Source leakage depth varied during the course of the experiment but without any significant relationship to the Soil Water Content changes or evaporative demand.' The current source leakage depth did not vary with transpirational demand or water content. But, didn't the spatial distribution of the leakage vary with transpiration demand and didn't you show that this was related to the occurrence of stress?

The reviewer is correct and this is even more apparent with the new figure 8b. Please refer to previous questions for more details on how this was described and discussed in the revised version of the manuscript.

---

## Author Response (AR2)

General comments
The authors have responded adequately to my previous comments. I still have a few remaining questions and remarks

*In the following our response to the reviewer's comment is written in blue.*

There is still a problem with figures 4 and 6 (and figures in the appendix) and the labeling of the cycles. The authors mix up the label numbering in the table, the figure and the text and then they even use two different numbering scales: one going from -1 to 8 and one going from 0 to 9.

*We carefully check that all the cycles starts at 0 and ends to 9. There was indeed one error, in the revised version L487 (corrected).*
*We found otherwise that all the figures are consistent (green= left irrigation, orange= right irrigation).*

The time lapse inversion is defined at line 354: '(2) time-lapse inversion (difference inversion) where the difference in resistivity is inverted between a given survey and a background survey (in this case, the background survey is the previous one). In this study, we used the second approach, which allowed filtering of systematic noise and highlights variations (as a percentage of differences) between two times.' Looking more in detail at the figure 5 and figures A2-A10, I got confused what was used now as background in the difference inversion. These figures seem to suggest that the background was the inversion just before the start of the irrigation in a cycle and that the same background was used for all difference inversions during a cycle.

*The reviewer is right. We used a background constrainst inversion. When we only had two times difference inversion and background inversion are similar. Sentence corrected.*

Blue areas would then be regions where the soil is wetted and red areas where it dries. This must be clarified as it also influences the interpretation of the difference inversions a lot.

*We agree with the reviewer suggestion and added a sentence: "The application of background constraint inversion, as illustrated in Figure 5bc, leads to an interpretation suggesting that the blue regions correspond to areas where the soil is wet, whereas the red regions correspond to areas where the soil is drying."*

Furthermore, it must be discussed whether the increases in resistivity that are observed just after an irrigation event are artefacts or not. It is hard to assume that root water uptake during such a short time after an irrigation event would lead to such large resistivity changes.

*We partially agree with the reviewer comment. Although the time elapsed after irrigation is relatively short, it's crucial to consider that the observed changes are relative to the background time, which exceeds 4 hours. Nonetheless, we have tempered our interpretation by incorporating the following statement: "Positive alterations in resistivity observed immediately after the irrigation event may potentially be artifacts stemming from strong gradient in resistivity*

*induced by the irrigation.*"

Figure 1 and its discussion: Make clear that for the wet soil, the water potentials are higher, so shift the potentials on the graph to the left.

*Ok done*

The purpose of the two lower diagrams should be to make clear that:
1) under dry soil conditions, the main part of the potential drop occurs in the soil-to-root part of the trajectory whereas in wet soil conditions, the main part of the potential drop is in the plant part of the trajectory. You could also include that when the transpiration rate is lower, that then the potential drop in the plant is lower than when the transpiration rate is higher. Your figure makes this clear.

*Sentence added "In dry soil conditions, the primary part of the potential drop happens within the soil-to-root connection, while in wet soil conditions, the main portion of the potential drop is in the plant section."*

2) when the soil is dry, the uptake is more uniform along the root system whereas when the soil is wet, the uptake is more from the parts in the root system where the resistance to flow to the collar is lower and these parts are closer to the root collar. This is not illustrated in this figure. Making the electrical analogue, I think you could do this by playing with the size of the resistor of the soil-to-root resistance. In wet conditions, you make it smaller than the resistors in the plant and in dry soil conditions, you make it much larger. Then it should be clear from the electrical analogue that the path from the upper soil to the collar experiences much less resistance than the path from the tip of the root. You could illustrate that by showing a larger flux Q in the top than in the bottom. For the dry conditions, the electrical analogue should show that the resistance to flow is similar for the top and the bottom (the soil resistance is the main component and the difference in plant resistance for top and bottom does not play such an important role anymore).
I suppose the difference between the electrical flow and the water flow is that the ratio of the resistance to flow in the root system for axial to radial flow is much larger for electrical current than for water flow. Plant tissues have evolved and adapted to have a small axial resistance to water flow. But, this does not mean that axial resistance to electric flow is similarly small.

*The reviewer's suggestion is interesting. Nevertheless we think that conveying the idea of the electrical analogue at a single root scale according to the soil conditions is a bit tricky. Just like in Couvreur 2012 Fig. 1, our conceptual figure only aimed to show the hydraulic analogue. At the current state of the art it is difficult to draw conclusion on the electrical one as we tried to demonstrate in this article.*

Detailed comments:

Ln 162: 'As the soil conductance $g_s$ is linked by the relationship between the transpiration rate

over the Δψsoil, for the same evaporation rate, gs is increasing when the soil dries out.' I am not following the reasoning here. The soil conductance should decrease when the soil dries out. Therefore, for a lower water content and the same transpiration, delta pis_soil should become larger (in absolute value).

The reviewer is right. Sentence rephrased.

Ln 164: 'For a constant soil conductance, when the evaporation rate is increasing the gs increase. The same occurs for the root conductance gr. The root axial water flow rates Qx (L3T−1) and root radial water flow rates Qr 165 (L3T−1) can be solved analytically by solving the system of equations of Ohm's and Kirchhoff's laws 166 (Couvreur et al., 2012)' Check this because I think this is not correct.

We removed part the first part of the sentence ("*For a constant soil conductance, when the evaporation rate is increasing the gs increase. The same occurs for the root conductance gr.*") as this statement needs much more carefull attention and we invite the reader to read details in previous works (Doussan et al., 1999, Manoli et al., 2014 and Couvreur et al., 2012 and Cai et al., 2022).

Ln 385: 'which is the same of the soil mixture in the rhizotron': Reformulate

Sentence rephrased: "*The rhizotron soil mixture porosity was assumed to be equal to 0.55*"

Ln 396: What is ‚partaerial'?

Corrected – we meant aerial part.

Ln 440: according to the numbering in table 1, in cycle number 3, water is still applied in all the holes on the left side.

Corrected (cycle 4 instead of 3).

Ln 453 figure 4: This figure is not yet fully consistent. In pannel a there are 10 bars corresponding with 10 applications. But, according to table 1, there should be 12 applications. The last two applications mentioned in table 1 are not included in the bar chart.

Figure 4 and table are now consistent.

The location of the bars on the x-axis do not match with the timing of the peaks in the bottom part. Therefore, the colors in the bar chart do not match with the colors in the chart below. For instance, the last bar is orange and the last peak is green. Second, the numbering of the cycles in the bar chart is not consistent with that in the table. For instance, the black bar with cycle number 5 in the figure actually corresponds to an application that is not numbered in the table.

Corrected.

Ln 479: The cycle numbers do not match with what is in the table or the figures.

Now the cycles number matches.

Probably the reviewer was confused because the table gives the **irrigation time** while the figures gives the **ERT survey time**. We improve the table legend to make it clear.
For figure 5 for instance:
Table 1 indicates that the **irrigation time** that starts cycle 7 is: 2022-06-29 13:45.
In Fig. 5, background time (a) 2022-06-29 9:30 is then located in cycle 6, while b and c are located in cycle 7 (as stated in the figure legend).

| (a)
Background (-4h) =
2022-06-29 9:30 | (b)
Just After Irrig. (+0h15)
= 2022-06-29 14:15 | (c)
6 days after Irrig =
2022-07-05 16:35 |
| --- | --- | --- |

Figure 5: Spatial distribution of the resistivity (in Ωm) and changes (in %) in ER obtained by a time-lapse inversion between cycle 6 and 7 following partial left irrigation of the rhizotron.

Ln 482: 'The soil CSD is not shown as it is always pinpointed to the location of the injection electrode' This sentence, I did not understand fully since in Figure 7, the soil CSDs are shown. I suppose you mean that the center of mass of the soil CSD is not shown?

Corrected thanks.

Ln 512: 'The most significant negative changes in averaged water content are attributable to the triggered irrigation, leading to a ΔΘ (change in water content) of -0.1. Conversely, positive changes primarily result from transpiration, with a maximum value located at +0.1.' This is confusing. I would associate a positive delta theta with an increase in theta and therefore would associate irrigation with a positive delta_theta and transpiration with a negative delta_theta. This is opposite to what is written here.

Corrected thanks.

Ln 552: may encourages should be may encourage.

Corrected.

Ln 567: There is a reference to figure 9 but there is no figure 9.

Corrected to fig. 8.